EMBO
Molecular Medicine

# Photoreceptor glucose metabolism determines normal retinal vascular growth

Zhongjie Fu[1] [ID], Chatarina A Löfqvist[2], Raffael Liegl[1], Zhongxiao Wang[1], Ye Sun[1], Yan Gong[1],
Chi-Hsiu Liu[1], Steven S Meng[1], Samuel B Burnim[1], Ivana Arellano[1], My T Chouinard[3], Rubi Duran[1],
Alexander Poblete[1], Steve S Cho[1], James D Akula[1], Michael Kinter[4], David Ley[5], Ingrid Hansen Pupp[5],
Saswata Talukdar[3], Ann Hellström[2] & Lois EH Smith[1],* [ID]

## Abstract

The neural cells and factors determining normal vascular growth
are not well defined even though vision-threatening neovessel
growth, a major cause of blindness in retinopathy of prematurity
(ROP) (and diabetic retinopathy), is driven by delayed normal
vascular growth. We here examined whether hyperglycemia and
low adiponectin (APN) levels delayed normal retinal vasculariza-
tion, driven primarily by dysregulated photoreceptor metabolism.
In premature infants, low APN levels correlated with hyper-
glycemia and delayed retinal vascular formation. Experimentally in
a neonatal mouse model of postnatal hyperglycemia modeling
early ROP, hyperglycemia caused photoreceptor dysfunction and
delayed neurovascular maturation associated with changes in the
APN pathway; recombinant mouse APN or APN receptor agonist
AdipoRon treatment normalized vascular growth. APN deficiency
decreased retinal mitochondrial metabolic enzyme levels particu-
larly in photoreceptors, suppressed retinal vascular development,
and decreased photoreceptor platelet-derived growth factor
(*Pdgfb*). APN pathway activation reversed these effects. Blockade
of mitochondrial respiration abolished AdipoRon-induced *Pdgfb*
increase in photoreceptors. Photoreceptor knockdown of *Pdgfb*
delayed retinal vascular formation. Stimulation of the APN path-
way might prevent hyperglycemia-associated retinal abnormalities
and suppress phase I ROP in premature infants.

**Keywords** adiponectin; hyperglycemia; metabolism; photoreceptor;
retinopathy of prematurity
**Subject Categories** Metabolism; Neuroscience; Vascular Biology &
Angiogenesis
2017 | Accepted 30 October 2017 | Published online 27 November 2017

## Introduction

About 15 million preterm infants are born yearly with an incom-
pletely vascularized retina (Hellstrom *et al*, 2013), which fails to
develop normally after birth as it would *in utero*. Failure of normal
neurovascular growth is the primary cause of later vision-threa-
tening pathologic angiogenesis driven by hypoxia and fuel insuffi-
ciency of the non-vascularized retina. It has been assumed that
supplemental oxygen is the major cause of delayed retinal vascular-
ization in preterm infants. However, fuel demand from retinal
neurons, particularly photoreceptors, is very high (Wong-Riley,
2010), and photoreceptor metabolic alterations can control patho-
logical angiogenesis (Joyal *et al*, 2016). We hypothesized that
photoreceptor fuel metabolism also controls normal vessel growth
(Fulton *et al*, 2009; Akula *et al*, 2010).

In very premature infants, hyperglycemia in the early postnatal
weeks strongly correlates with later retinal neurovascular develop-
ment (Garg *et al*, 2003; Ertl *et al*, 2006; Chavez-Valdez *et al*, 2011;
Kaempf *et al*, 2011; Mohamed *et al*, 2013; Ahmadpour-Kacho *et al*,
2014; Mohsen *et al*, 2014; Au *et al*, 2015). It is unknown whether or
how hyperglycemia suppresses retinal vessel growth. Postnatal
hyperglycemia occurs in ~80% of preterm infants born weighing
< 750 g and ~45% of preterm infants born < 1,000 g (Dungan *et al*,
2009), but the cause and results of hyperglycemia are not well
understood. Exogenous insulin does not adequately regulate hyper-
glycemia nor prevent hepatic gluconeogenesis in these infants
(Chacko *et al*, 2011). Insulin resistance and defective islet β-cell
processing of pro-insulin probably influences preterm hyper-
glycemia (Mitanchez-Mokhtari *et al*, 2004). We hypothesized that
the metabolic modulator APN is involved (Scherer *et al*, 1995;
Oberthuer *et al*, 2012). As low serum APN in preterm infants
predicts later pathological angiogenesis (Fu *et al*, 2015), we hypoth-
esized that APN modifies the suppressive effects of hyperglycemia
on neuronal metabolism to control normal vascularization. With the

1   Department of Ophthalmology, Boston Children's Hospital, Harvard Medical School, Boston, MA, USA
2   Section for Ophthalmology, Department of Clinical Neuroscience and Rehabilitation, Institute of Neuroscience and Physiology, Sahlgrenska Academy, University of
    Gothenburg, Göteborg, Sweden
3   Merck Research Laboratories, Boston, MA, USA
4   Aging and Metabolism Research Program, Oklahoma Medical Research Foundation, Oklahoma City, OK, USA
5   Pediatrics, Department of Clinical Sciences, Skåne University Hospital and University of Lund, Lund, Sweden
    *Corresponding author. Tel: +1 617 919 2529; E-mail: lois.smith@childrens.harvard.edu

## Results

### Hyperglycemia correlated with delayed retinal vascular development and with low APN levels in premature infants

We correlated hyperglycemia with serum APN concentrations (divided into tertiles) in a prospective longitudinal study of 50 preterm infants. Mean plasma glucose levels were lower in higher APN tertiles in postnatal weeks 1 and 2 (Fig 1A). APN levels increased in all premature infants during the early postnatal period (Fig 1B) although preterm infants with delayed retinal vascularization had lower APN levels than those with normal retinal vascularization (Fig 1B and C). This suggested that hyperglycemia correlated with suppressed normal retinal vascularization and APN might counteract the impact of hyperglycemia and delayed neurovascular retinal development.

### Induction of hyperglycemia delayed retinal neurovascular maturation in mice

To clarify the influence of hyperglycemia and APN on retinal neural development, particularly photoreceptors, and on retinal vascularization in the early postnatal period, we established a neonatal mouse model of hyperglycemia-associated retinopathy (HAR) (phase I ROP comprising poor vessel development) (Fig 2). In human and mouse retina, blood vessels grow from the central to the peripheral retina in three layers. In mouse, the superficial layer forms at postnatal (P) days 1–10, the deep layer at P8-P12, and the intermediate layer at P14-P20 (Fig 2A; Dorrell & Friedlander, 2006). In the model of hyperglycemic-associated retinopathy, we injected streptozotocin (STZ, 50 $mg^{-1}$ $kg^{-1}$ $day^{-1}$) intraperitoneally into C57BL/6J (wild type; WT) mice from P1-P9 (Fig 2A). Hyperglycemic versus normoglycemic retinas had suppressed deep vascularization with fewer vascular meshes, reduced vessel length, and reduced vascular area (Figs 2B and C, and EV1A). The body weight of hyperglycemic and control mice was comparable (Fig EV1B), suggesting that the delay in retinal vascularization was not secondary to general growth retardation. Hyperglycemia was induced at P8 (Fig EV1C). As expected, STZ-treated mice had distorted pancreatic islet cells (Fig EV1D). Notably, the vascular developmental delay was found only in the deep layer (Fig EV1E), which is closest to photoreceptors, consistent with our hypothesis, that the high metabolic demands of photoreceptors might direct vessel growth. Insulin treatment rescued the deep layer vessel growth (Fig EV1F), further confirming that hyperglycemia induced retinal vascular abnormality. Elevated blood glucose and triglyceride levels and reduced insulin levels were found at P10 (Fig 2D). No significant changes in triglyceride levels were observed at P8 (Fig EV1C). Intravitreal STZ injections had no direct effect on retinal vascularization (Fig EV2A), suggesting that delayed vessel growth was secondary to hyperglycemia and/or insulin deficiency.

### The APN pathway modulated retinal vascular development in hyperglycemic retinopathy

We then assessed whether APN modulated HAR. In hyperglycemic mice, there was a 1.5-fold increase in serum APN levels, while retinal *Apn* levels were low and comparable (Fig 3A). In retinal

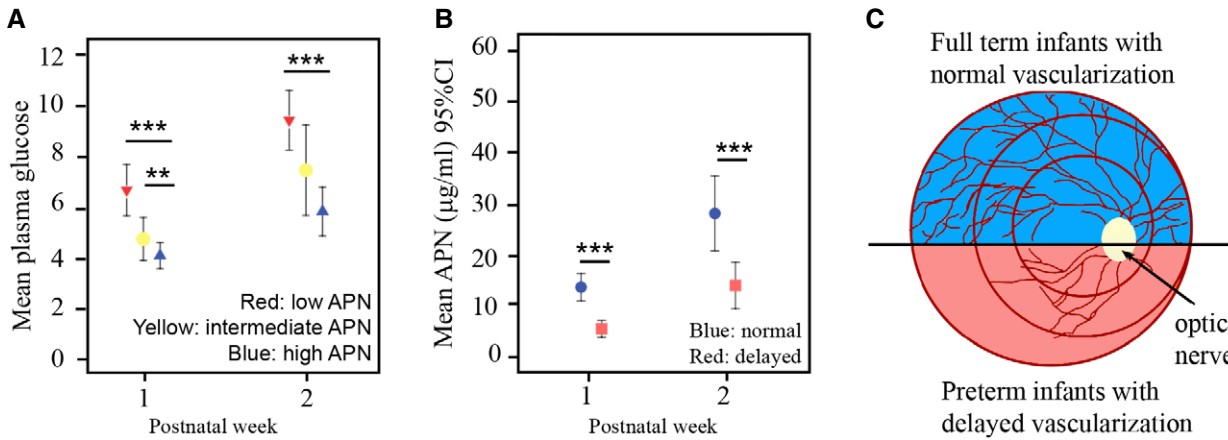

**Figure 1. Hyperglycemia positively correlated with low serum APN levels and low APN levels positively correlated with delayed retinal vascularization in premature infants (gestational age at birth: 23–30 weeks, birth weight: 348–1,716 g).**

A   Mean weekly APN levels and 95% confidence interval for mean plasma glucose concentration in relation to APN level tertiles (T). At postnatal weeks 1 and 2, infants in the $T_{low}$ APN had significantly higher mean plasma glucose concentrations than infants in the $T_{high}$. APN $T_{low}$ 3.86 (range: 1.79–5.97) µg $ml^{-1}$ (red), APN $T_{intermediate}$ 8.63 (range: 6.43–11.54) µg $ml^{-1}$ (yellow), and APN $T_{High}$ 17.50 (11.84–36.00) µg $ml^{-1}$ (green). $n$ = 50, **$P$ < 0.01; ***$P$ < 0.001. $t$-test.

B   Mean weekly APN levels and 95% confidence interval in infants with normal (blue) versus delayed (red) vascularization. $n$ = 44, ***$P$ < 0.001. $t$-test.

C   Schematics of retinal vascularization in premature versus term infants. Preterm (red) versus term (blue) infants had delayed retinal vascularization. Circles in red represent zones of retina in preterm infants.

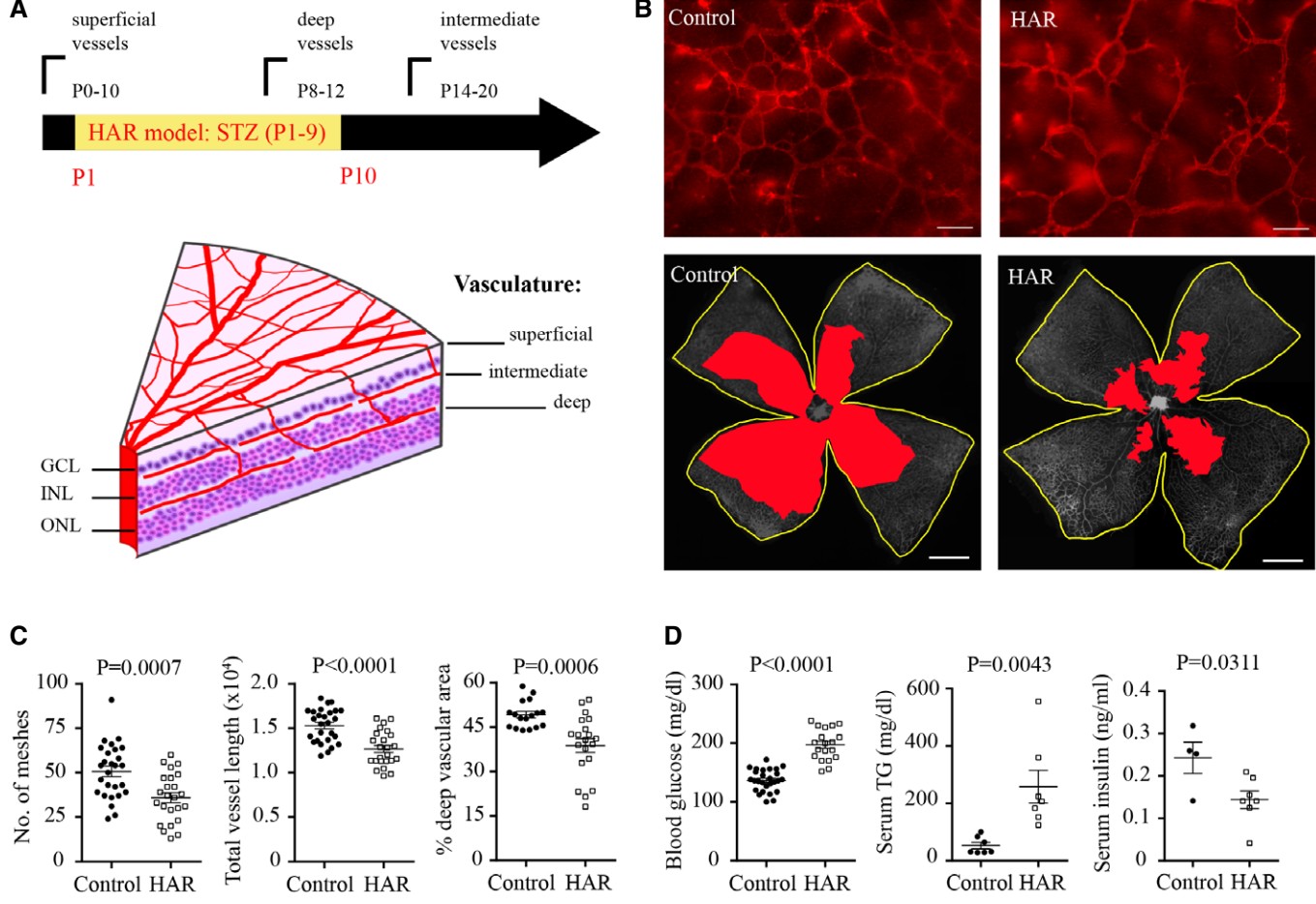

**Figure 2.   A novel mouse model of hyperglycemia-associated retinopathy (HAR) was established and compared with normoglycemic controls in P10 neonates..**

A   The mouse neonates received intraperitoneal injection of STZ and schematic graph shows mouse retinal vessel development after birth (left). GCL, ganglion cell layer; INL, inner nuclear layer; ONL, outer nuclear layer.

B   Representative images of the deep vascular network in isolectin-stained (red, top) retinal whole mounts of HAR and controls. Scale bar, 50 μm (top) and 1 mm (bottom). The total retinal area is outlined in yellow and deep vascular coverage is highlighted in red (bottom, grayscale).

C   Quantification of deep vascular network of HAR and controls. $n = 17$–27 retinas/group. Data presented as mean $\pm$ SEM, unpaired $t$-test.

D   Blood glucose ($n = 19$–27 retinas/group), serum insulin (ELISA) ($n = 4$–7 pooled samples/group) and triglyceride (TG) concentrations ($n = 7$/group) of HAR and controls. Data presented as mean $\pm$ SEM, unpaired $t$-test.

Data information: See also Fig EV1.

---

neurons and vessels isolated with laser microdissection, APN receptor 1 mRNA *(AdipoR1)* was highly expressed in photoreceptors (ONL) and *AdipoR1* was increased in HAR (Fig 3B).

To evaluate the influence of APN on ocular vessels in normoglycemic and hyperglycemic mice, we first evaluated hyaloid vessel regression (Fig 3C). In both mice and human eyes, the hyaloid vessels are the first to form *in utero* and regress as the retinal vessels develop (Ito & Yoshioka, 1999; Stahl *et al*, 2010). In humans, hyaloid vessels normally regress before term birth but are present at birth in preterm infant eyes (Jones, 1963); chronically persistent hyaloid remnants are associated with delayed retinal vascular development and visual dysfunction (Sharma & Biswas, 2012). We found that hyperglycemia and APN deficiency led to delayed regression of the hyaloid vessels at P8 (Fig 3C). Hyaloid persistence was greater in $Apn^{-/-}$ eyes (35 of 37; 94.5%) versus WT eyes (31 of 48; 64.5%)

at P30 (Fig 3D), suggesting that hyperglycemia and low APN might lead to hyaloid persistence in premature infants.

Given that APN deficiency delayed hyaloid regression, we then examined the effect of APN deficiency on normal retinal vascularization. APN deficiency attenuated superficial but not deep retinal vascularization in normal condition at P10 (Fig EV2B). In hyperglycemic $Apn^{-/-}$ versus hyperglycemic WT mice (mean glucose $Apn^{-/-}$: $206 \pm 8$ mg dl$^{-1}$; WT: $197 \pm 6$ mg dl$^{-1}$), deep vascular formation was suppressed (Figs 3E and EV2C). To evaluate whether activation of the APN pathway could normalize retinal vascularization, we treated WT hyperglycemic mice with recombinant mouse APN or APN receptor 1 and 2 dual agonist, AdipoRon (Okada-Iwabu *et al*, 2013) (administrated from P7 to P9 when the deep layer starts to form). Both reversed the vascular developmental suppression (Figs 3F and G, and EV2D and E). We speculated that APN

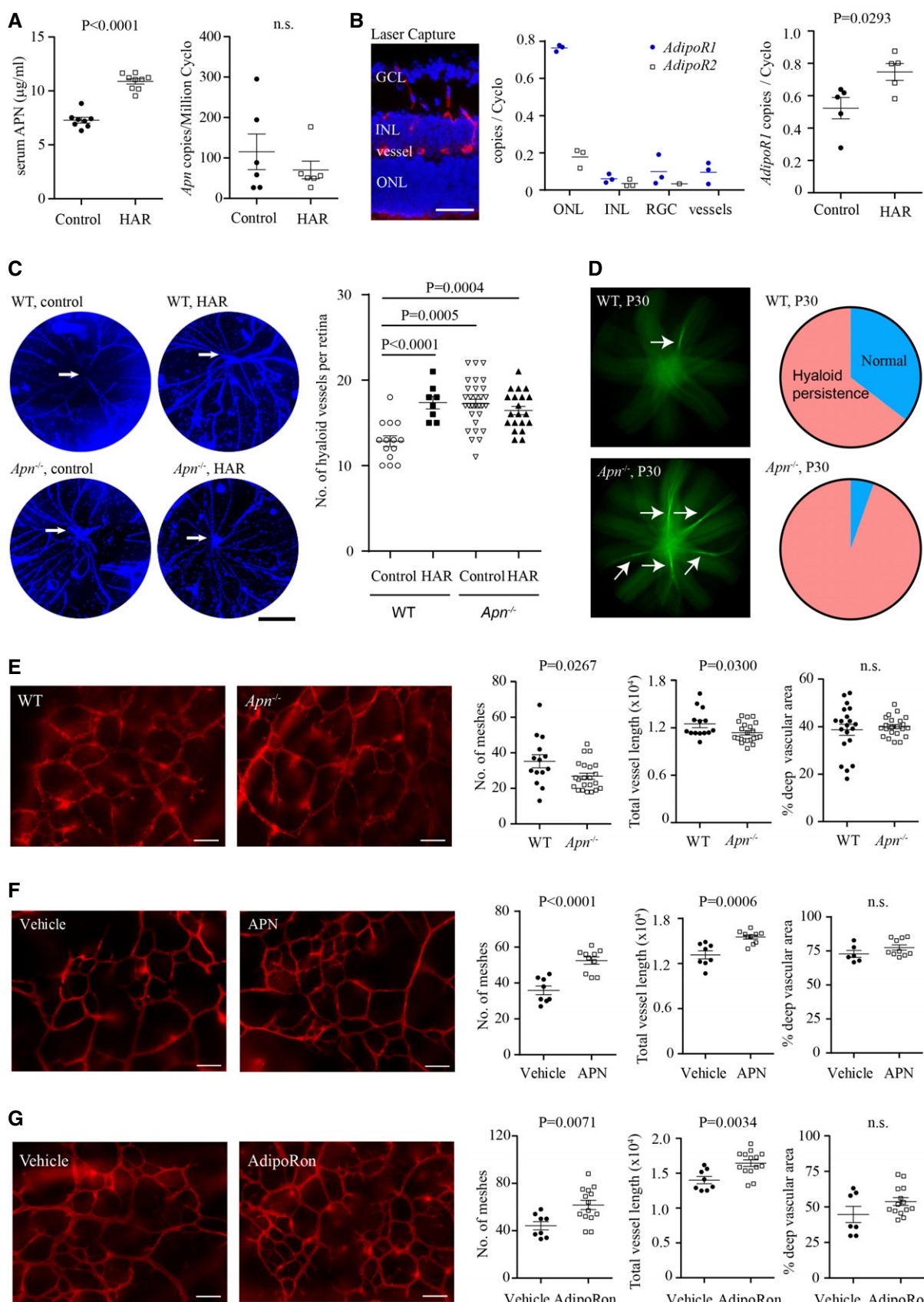

**Figure 3.**

◀

**Figure 3.  APN pathway activation protected against hyperglycemia-associated retinopathy (HAR) delay in P10 neonatal mice.**

A    Left: serum APN levels (ELISA) ($n = 8$–9 retinas/group); right: retinal *apn* (qRT–PCR) ($n = 6$ retinas/group) of HAR and controls.
B    Left: retinal cross-sectional layers for laser capture microdissection (LCM: DAPI for nuclei, blue; isolectin for vessels, red); center: mRNA levels of *AdipoR1* and *AdipoR2* in retinal neuronal layers and vessels. $n = 3$ pooled retinas/group. Right: *AdipoR1* mRNA in HAR and control retinas. $n = 5$ retinas/group.
C    Control and HAR eyes in WT and $Apn^{-/-}$ mice. Left: representative images of DAPI-stained hyaloid vessels (blue). Right: quantification of preserved hyaloid vessels branching from the hyaloid artery (white arrow). Scale bar, 1 mm. $n = 8$–30 retinas/group.
D    Left: Fundus photograph focused to show persistent hyaloid vessels (white arrows) (green, fluorescein AK-FLUOR) in WT and $Apn^{-/-}$ mice at P30 ($n = 37$–48 retinas/group); right: pie graph of percentage of eyes examined with persisting hyaloid.
E    Left: In WT and $Apn^{-/-}$ representative images of deep retinal vasculature (lectin, red) in whole-mounted retinas; right: quantification of deep vessels ($n = 14$–22 retinas/group).
F, G    Left: Representative images of deep retinal vasculature (lectin, red) in whole-mounted retinas of WT hyperglycemic (HAR) mice with recombinant mouse APN treatment (F) ($n = 6$–10 retinas/group) or with AdipoRon treatment (G). Right: quantification of deep retinal vasculature ($n = 7$–14 retinas/group).

Data information: Scale bars, 50 μm (B, E–G) or 1 mm (C). Data presented as mean ± SEM; unpaired *t*-test (A, B, E–G) or ANOVA (C). See also Fig EV2.

activation of AdipoR1 in photoreceptors might control vascularization as *AdipoR1* was the most highly expressed APN receptor in mouse retina and was localized mainly in photoreceptors (Fig 3B).

### The APN pathway regulated retinal metabolism and photoreceptor neurovascular growth factor production

Given that APN is a key metabolic modulator, we examined APN's effect on retinal metabolic enzyme concentrations in hyperglycemic-associated retinopathy with quantitative proteomics. APN deficiency decreased the level of enzymes involved in glucose metabolism, the Krebs cycle and the electron transport chain (Fig 4A), suggesting a reduction in metabolism and energy production with APN deficiency. In retinal punches, decreased the oxygen consumption rate (OCR, an indicator of mitochondrial respiration) and extracellular acidification rate (ECAR, an indicator of glycolysis) was seen in hyperglycemic retina (Fig EV3A); msAPN increased OCR (Fig 4B) and ECAR (Fig EV3F) in WT HAR retinas. Metabolic enzyme *Hk1* and *Cs* mRNA levels were reduced in photoreceptors (ONL, Fig 4C) but not in vessels (Fig EV3B), suggesting that the leading contribution to metabolic alterations in hyperglycemic $Apn^{-/-}$ retinas was photoreceptors.

We confirmed our findings in photoreceptors *in vitro*. In immortalized cone photoreceptor cells (661W), AdipoRon induced phosphorylation of AMP-activated protein kinase (AMPK, a key mediator of APN in modulating mitochondrial function in diabetes (Iwabu *et al*, 2010; Fig 4D) and increased *Hk1* and *Cs* levels (Fig 4E). AdipoRon reversed hyperglycemia-induced disruption of mitochondrial morphology and increased mitochondrial fusion protein production (Fig EV3C–E), suggesting that AdipoRon prevented mitochondrial fission, an early event in cell death (Knott *et al*, 2008). AdipoRon also increased OCR and ECAR (Figs 4F and EV3G), confirming that the APN pathway promotes photoreceptor oxygen and glucose use. Inhibition of AMPK with Compound C (McCullough *et al*, 2005; Liu *et al*, 2014) attenuated the protective effects of msAPN on retinal vascular development in HAR (Fig EV3H), further suggesting AMPK is an essential mediator in APN pathway. Finally, AdipoRon increased platelet-derived growth factor B (*Pdgfb*), which promotes retinal vascular growth in mice (Lindblom *et al*, 2003). Inhibition of adenosine triphosphate synthase with oligomycin abolished the AdipoRon induction of *Pdgfb* in 661W cells (Fig 4G). msAPN administration increased retinal expression of *Pdgfb* (Fig 4H). *Pdgfb* expression was decreased in photoreceptors of $Apn^{-/-}$ versus WT mice (Fig 4I). Photoreceptor-specific knockdown

of *Pdgfb* inhibited retinal vascular network formation in WT mice (Fig 4J and K). APN deficiency decreased *Vegfa* expression in the ONL while photoreceptor-specific knockdown of *Vegfa* did not change retinal vascular formation in WT mice (Fig EV4A and B).

### APN restored hyperglycemia-suppressed retinal function

We obtained electroretinographic (ERG) recordings in hyperglycemic mice. Mathematical modeling of the photoreceptor response, captured in the leading edge of the electroretinographic *a*-wave (Hood & Birch, 1994), detected change in photoreceptor amplitude ($Rm_{P3}$) and in post-receptor response sensitivity ($\sigma$) and amplitude ($V_m$). This resulted in a decrease in retinal sensitivity ($S_m$) (Akula *et al*, 2008; Fig 5A). The changes in post-receptor sensitivity correlated with the sum of deficits in photoreceptor sensitivity and saturated amplitude (Fig 5B), seen in premature infants (Hansen *et al*, 2016). Decreased thickness of the photoreceptor layer was observed in hyperglycemic retinas (Fig 5C). Long-term administration of mouse recombinant APN (msAPN) rescued post-receptor responses ($V_m$) in rod and cone ERG (Fig 5D and E), reflecting better photoreceptor function.

In summary, APN pathway activation improved photoreceptor function during hyperglycemic stress, by increasing mitochondrial activity and inducing neurovascular growth factor production (PDGFB, not VEGFA, Fig 6) in photoreceptors, which improved normal retinal vascularization.

## Discussion

We found that in premature infants, hyperglycemia was associated with delayed retinal vascularization, and correlated with low circulating APN levels. In our mouse model of HAR, activation of the APN pathway reversed hyperglycemia-associated suppression of photoreceptor metabolism and increased the consumption of oxygen and glucose, stimulated neurovascular growth factor (PDGF) production, and normalized vascularization necessary for oxygen and nutrient supply. APN deficiency worsened hyperglycemia-associated suppression of retinal neurovascular maturation (Fig 5).

In premature infants, early hyperglycemia coinciding with phase I of ROP contributes to the later progression of the neovascular phase of ROP (Garg *et al*, 2003; Ertl *et al*, 2006; Chavez-Valdez *et al*, 2011; Kaempf *et al*, 2011; Mohamed *et al*, 2013; Ahmadpour-Kacho *et al*, 2014; Mohsen *et al*, 2014; Au *et al*, 2015). We

    

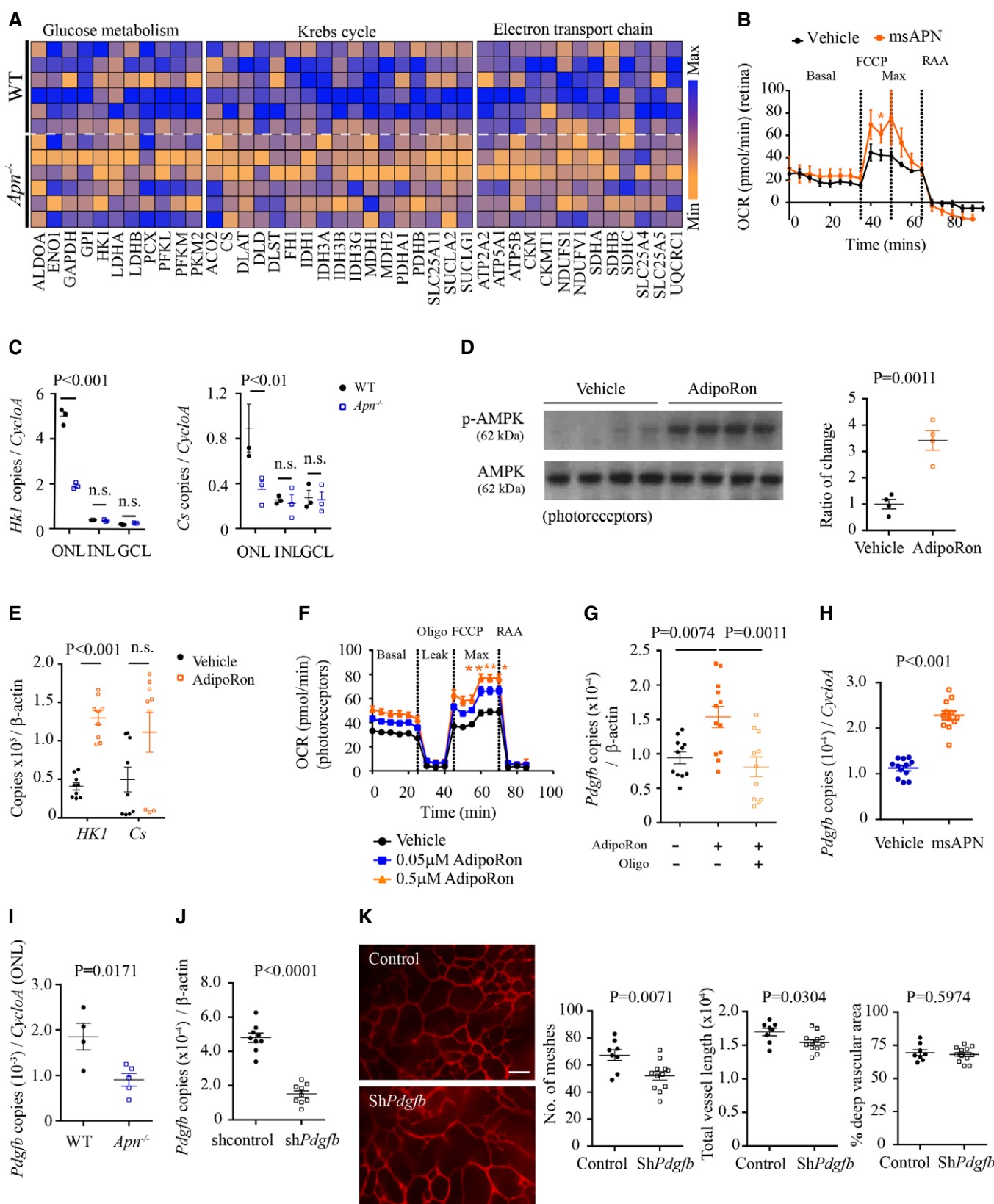

**Figure 4.**

◀

**Figure 4.   APN pathway activation modulated photoreceptor mitochondrial function in hyperglycemic mice.**

A   Quantitative proteomic analysis of key glucose metabolic enzymes and mitochondrial metabolic enzymes in hyperglycemic (HAR) WT and $Apn^{-/-}$ mice at P10. $n = 6$ mice/group. Blue: maximum of levels; orange: minimum of levels.

B   Seahorse XF$^e$96 Flux Analyzer oxygen consumption rate (OCR) of WT HAR P10 retinas treated with msAPN or vehicle (PBS). Basal: initial OCR without treatment; max, maximal OCR: after addition of FCCP (carbonyl cyanide-4-trifluoromethoxy phenylhydrazone). Treatment with rotenone and antimycin A (RAA) reveals non-mitochondrial respiration. ANOVA, *$P < 0.05$. $n = 7$–8 per group.

C   mRNA expression of metabolic enzyme ($Hk1$, $Cs$) in retinal neuronal layers (ONL, INL, GCL) of $Apn^{-/-}$ versus WT HAR P10 retinas (LCM and qRT–PCR). $n = 3$ pooled retinas/group. ANOVA. n.s., not significant.

D   Western blot for p-AMPK/AMPK in 661W cells treated with AdipoRon or vehicle. $n = 4$ per group. Unpaired $t$-test.

E   qRT–PCR of key metabolic enzymes ($Hk1$, $Cs$) in AdipoRon- versus vehicle-treated 661W cells. $n = 9$ per group. Unpaired $t$-test. n.s., not significant.

F   OCR of 661W cells treated with AdipoRon or vehicle. Leak indicates OCR independent of ATP production with proton leak induced with oligomycin (oligo) treatment. $n = 10$ per group. ANOVA. *$P < 0.05$, **$P < 0.01$.

G   qRT–PCR of $Pdgfb$ in AdipoRon- or vehicle-treated 661W cells with and without oligomycin. $n = 12$ replicates/group. ANOVA.

H   qRT–PCR of $Pdgfb$ in msAPN- or vehicle-treated WT HAR retinas. $n = 12$ replicates/group. Unpaired $t$-test.

I   $Pdgfb$ expression in photoreceptors (ONL from LCM) from $Apn^{-/-}$ versus WT HAR P10 mice (LCM and qRT–PCR). $n = 4$–5 replicates from three pooled retinas/group. Unpaired $t$-test.

J   qRT–PCR of $Pdgfb$ in hRK-sh$Pdgfb$-transfected 661W cells. $n = 9$ per group. Unpaired $t$-test.

K   Representative images of deep retinal vascular network and quantification in AAV2-hRK-Pdgfb-GFP versus AAV2-hRK-GFP (control) virus subretinally injected in WT mice. Retinas were examined at P10. Scale bar, 50 μm. $n = 8$–12 retinas/group. Unpaired $t$-test.

Data information: Data presented as mean ± SEM. See also Figs EV3 and EV4.

previously reported that low APN levels positively correlate with retinal neovascularization in premature infants and APN mediates the use of essential long-chain polyunsaturated fatty acids omega-3 in the retina to prevent pathological angiogenesis in mice (Fu *et al*, 2015). In adult patients, elevated serum APN levels in type 1 diabetes positively correlate with insulin sensitivity (Pereira *et al*, 2012) and plasma total antioxidant status (Prior *et al*, 2011). Serum APN deficiency may contribute to insulin resistance in diabetes (Iwabu *et al*, 2010) and an APN receptor 1 and 2 agonist reduces plasma glucose in diabetic mice (Okada-Iwabu *et al*, 2013), suggesting that APN may protect against hyperglycemia-associated metabolic complications including retinal alterations. This suggestion is supported by our results, which demonstrate that with low levels or deficiency of APN, there is worse retinal vasculature development in HAR.

Some reports suggest that vascular metabolism controls endothelial cell growth (De Bock *et al*, 2013; Schoors *et al*, 2015). However, we found that photoreceptor metabolism also controls blood vessel growth. Retina (in particular photoreceptors) is the most metabolically demanding tissue in the body (Wong-Riley, 2010) and photoreceptors have the highest number of mitochondria of any cell (Hoang *et al*, 2002). Early photoreceptor dysfunction predicts subsequent pathological angiogenesis in neonatal rats (Akula *et al*, 2007, 2010). Photoreceptor oxidative stress and inflammation disturbs the stability of blood vessels in early diabetic retinopathy (Du *et al*, 2013; Tonade *et al*, 2016). Other lines of evidence also show that AdipoR1 deficiency in mice is associated with decreased energy metabolism (Bjursell *et al*, 2007) and loss of AdipoR1 leads to mouse photoreceptor degeneration (Rice *et al*, 2015). A gene mutation in *ADIPOR1* causes human retinitis pigmentosa (with photoreceptor loss resulting in blindness) (Xu *et al*, 2016). These reports suggest that the APN pathway may play an important role in maintaining photoreceptor metabolic homeostasis. Here, we found that there was decreased metabolism in hyperglycemic retinas which improved with activation of the APN pathway. Pharmaceutical administration of APN protected against hyperglycemia-dampened rod and cone function, which was reflected in post-receptor ERG responses. Photoreceptors control developmental vascularization in the

retina and the APN pathway positively regulates photoreceptor metabolism.

The hyperglycemia status was not further changed with APN deficiency, suggesting that the impact of APN on retinal vascular growth was not through the adjustment of circulating glucose levels. Instead, APN directly regulated local retinal metabolism particularly photoreceptors. APN deficiency decreased key metabolic enzymes in hyperglycemic retinas, suggesting that stimulation of the APN pathway in hyperglycemic mice promoted retinal cell mitochondrial activity and glycolysis, which could compensate for the decreased retinal metabolism due to the lack of insulin. Further activation of the APN pathway with administration of recombinant APN or with administration of the APN receptor agonist AdipoRon improved retinal vascular formation through increased retinal and photoreceptor metabolic activity.

In both normoglycemic and hyperglycemic retinas, APN deficiency induced hyaloid vessel persistence, indicating that APN modulation of hyaloid vessel regression occurred independent of glycemic status. Retinal blood vessels normally branch from the superficial layer to form the deep vasculature as the hyaloid vessels regress, suggesting that persistent hyaloid vessels which supply oxygen and nutrients to the inner retinal cells may contribute to the reduced blood vessel network in the superficial retinal layer. Under hyperglycemic conditions, APN deficiency worsened the photoreceptor metabolic dysfunction and further decreased the deep vascular formation. These observations suggest that APN may work through different mechanisms to modulate hyaloid vessel regression, superficial and deep retinal vascular formation. There is a sharp decrease in the hyaloid vessel counts from P5 to P8 in mice (Rao *et al*, 2013). In HAR, hyperglycemia was induced from P6 to P8. The rise in blood glucose levels may interrupt hyaloid vessel regression during this sensitive period, which leads to persistent hyaloid vessels. However, as the superficial retinal vessels cover more than 50% of the retinas at P5 and almost reach the most peripheral region at P8, the impact of hyperglycemia on superficial vasculature may be mild. In $Apn^{-/-}$ mice, there is an early influence of APN loss on hyaloid and retinal vessels. Further investigation of the influence of APN pathway on the systems controlling

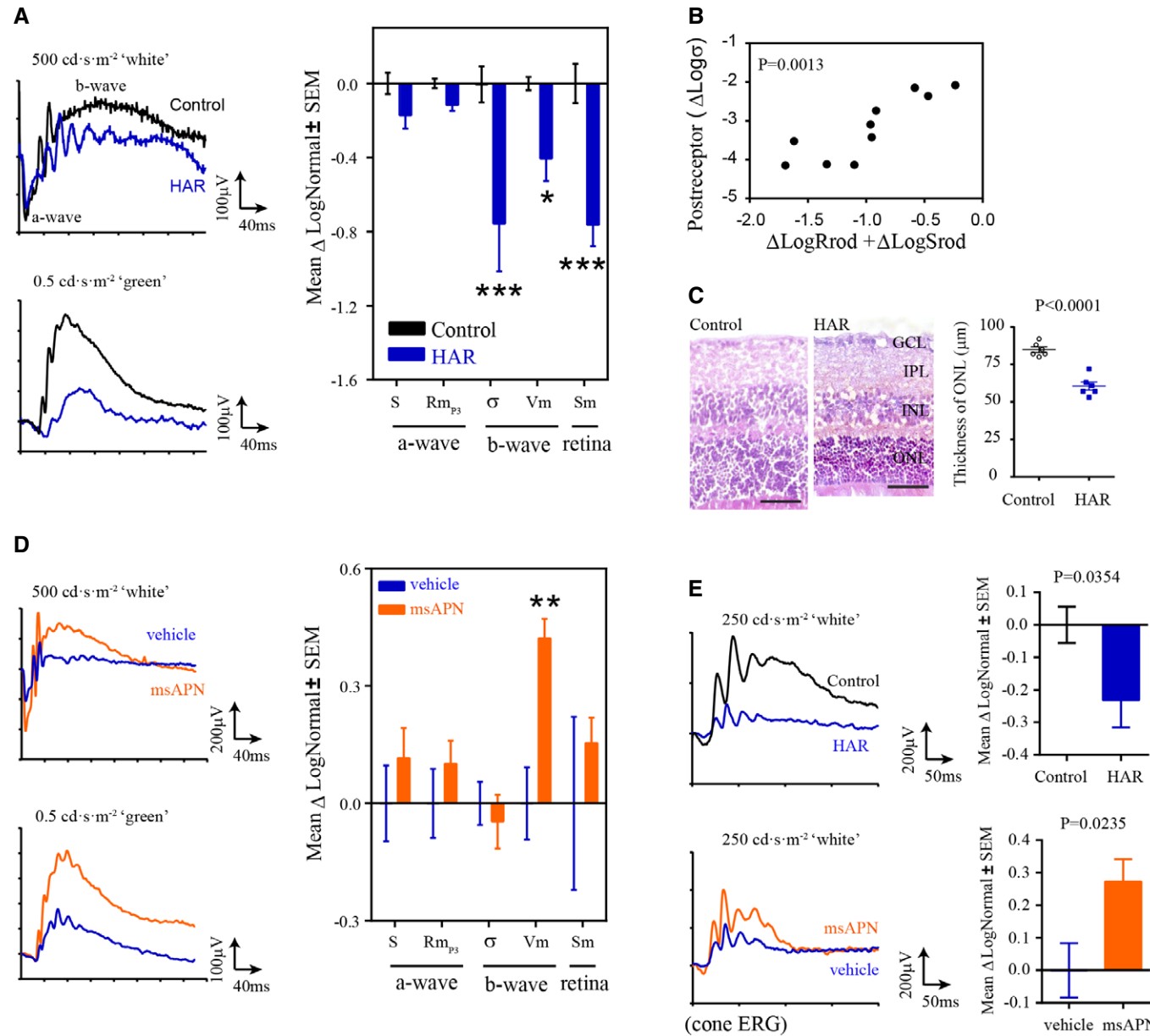

**Figure 5.  Recombinant APN treatment restored retinal function in hyperglycemia-associated retinopathy.**

A    Electroretinogram (ERG) assessment of retinal function in control and HAR mice showing photoreceptor response amplitude ($Rm_{P3}$), sensitivity (S), and post-receptor response amplitudes ($V_m$), sensitivity ($\sigma$), and retinal sensitivity ($S_m$). ERG parameters are presented as the log change from control mice. Representative ERG waveforms (left). $n = 10–12$ per group. *$P < 0.05$, ***$P < 0.001$. ANOVA.

B    Correlation of post-receptor sensitivity deficits ($Log\sigma$) with the sum of deficits in photoreceptor sensitivity ($LogSrod$) and saturated amplitude ($LogRrod$). $n = 10–12$ per group. Pearson $R$ test.

C    Measurement of outer nuclear layer thickness in WT control and HAR retinas at P30. Representative images of H&E-stained retinal cross sections (left). Scale bar, 50 μm. GCL, ganglion cell layer; IPL, inner plexiform layer; INL, inner nuclear layer; ONL, outer nuclear layer. $n = 6$ per group. Unpaired $t$-test.

D, E    Hyperglycemia was induced with 25 mg kg$^{-1}$ STZ (i.p., daily from P2 to P12) in WT mice. The mouse pups received msAPN (0.6 μg g$^{-1}$) or vehicle treatment daily from P7 to P39. Littermate controls were used. At P40, rod (D) and cone (E) ERG were conducted. Cone ERG was reflected by post-receptor responses ($V_m$). $n = 7$ to 12 per group. **$P < 0.001$. ANOVA (D) or $t$-test (E).

Data information: Data presented as mean ± SEM (A, C–E).

hyaloid vessel regression and superficial retinal vascular formation [e.g., retinal ganglion cells (Rao *et al*, 2013)] is needed to better understand mechanisms. APN receptors were expressed in retinal ganglion cells, although the expression was much less than in photoreceptors. In summary, APN, possibly through targeting different retinal neurons, controls retinal vascular formation. Further

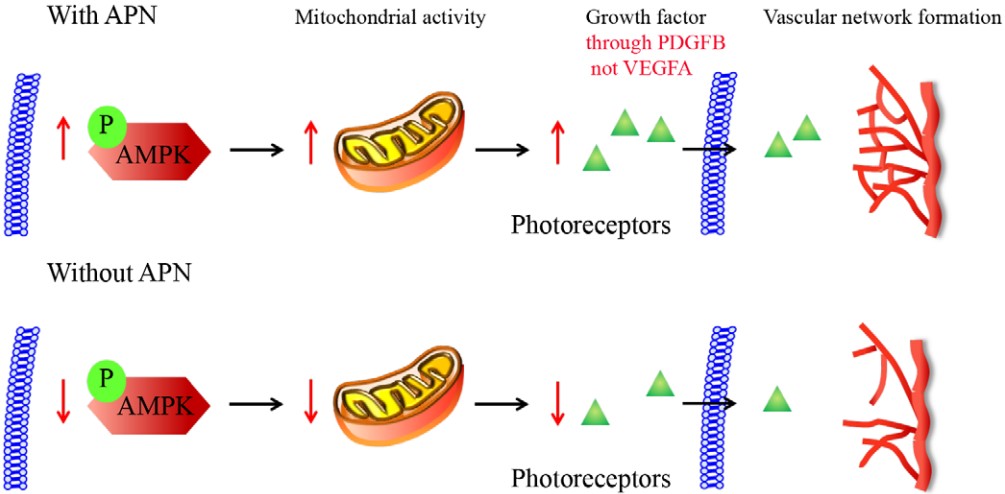

**Figure 6.  Schematic of proposed APN pathway in photoreceptor metabolism and deep retinal vascular development in STZ-induced hyperglycemic retinopathy.**

The APN pathway phosphorylates AMPK to increase mitochondrial activity, which in turn increases the production of neurovascular growth factor PDGFB. Photoreceptor-derived PDGFB (not VEGFA) promotes the deep retinal vascular network formation in hyperglycemic retinopathy.

exploration may greatly benefit patients with persistent hyaloid remnants (Mullner-Eidenbock *et al*, 2004; Saint-Geniez & D'Amore, 2004; Shastry, 2009; Sharma & Biswas, 2012).

There are some limitations to our study. First, the neonatal mouse model of STZ-induced HAR may not accurately mimic the hyperglycemic status in premature infants with insulin deficiency and insulin resistance (Mitanchez-Mokhtari *et al*, 2004). Second, the mouse retina does not have a macula, rich in cone photoreceptors as seen in primates. Third, further study is needed to define how photoreceptor metabolism influences vascular growth factor production and to determine what factors are modulated by APN to facilitate deep vascular growth. Although we demonstrate that PDGFB not VEGFA are involved in the process, other factors may also control blood vessel growth.

In summary, our data highlight the importance of hyperglycemia and its influence on photoreceptor metabolism that mediates early retinal neurovascular developmental delay. Targeting photoreceptor metabolism may help prevent hyperglycemia-associated retinal diseases like retinopathy of prematurity (ROP) and perhaps diabetic retinopathy. APN is a potential therapeutic target in retinal metabolic diseases.

# Materials and Methods

### Study design

This translational study aimed to explore the impact of hyperglycemia on retinal vascular development in premature infants and the role of APN on neural retinal metabolism. In premature infants, we correlated blood APN levels with blood glucose levels and retinal vascularization. We then established a neonatal mouse model of HAR to investigate APN control of neurovascular growth. In this model, we evaluated retinal vascular development with APN deficiency, or pharmacological supplementation of recombinant APN or

activation of APN receptors; the signaling pathways were also evaluated.

### Quantitative analyses of plasma glucose

Blood sampling for plasma glucose was performed from an umbilical arterial catheter or a peripheral arterial line, and plasma glucose was analyzed immediately after sampling by a blood gas analyzer (ABL 735; Radiometer, Copenhagen, Denmark). In total, 2,128 arterial samples were obtained and analyzed for plasma glucose during postnatal days (PND) 1–21.

### Blood sampling and quantitative APN analyses

Blood samples were obtained from umbilical cord blood at birth and from neonatal blood prior to enteral feeding at 72 h of age, at 7 days postnatally, weekly until at least a gestational age 35 weeks and again at a term age (postmenstrual age 40 weeks). Sampling was initially performed from an umbilical or peripheral arterial catheter and later by venous puncture. After centrifugation, serum samples were stored at −80°C until assayed.

All serum samples were diluted 1:306 and APN levels assayed using a human adiponectin ELISA kit (E091M, Mediagnost, Reutlingen, Germany). The intra-assay coefficients of variation were 3.8% at 3.9 μg ml$^{-1}$ and 4.7% at 13.1 μg ml$^{-1}$; the inter-assay coefficient of variation was 16.3% at 9.9 μg ml$^{-1}$. For values > 80 μg ml$^{-1}$, samples were further diluted using assay diluent and the assay repeated so that the results fell within the range. Samples were corrected for respective dilution.

### Examination of retinal vascular growth in premature infants

Examination of the retinal vascular growth started at 5–6 weeks of age but not before 31 weeks PMA. Retinal examinations through dilated pupils were performed biweekly to twice a week depending

on the extension of delayed retinal vascular growth, until the retina was fully vascularized.

## Neonatal mouse model of hyperglycemia-associated retinopathy

The streptozotocin (STZ)-induced hyperglycemia in neonatal C57BL/6J mice was adapted from previous studies (Cox *et al*, 2010; Kermorvant-Duchemin *et al*, 2013). For each experiment, aliquots of STZ were pre-weighed and kept in the dark at $-20°C$. STZ was dissolved in phosphate-buffered saline (PBS) immediately before injection to a final concentration of 50 $mg^{-1}$ $ml^{-1}$. 50 $mg^{-1}$ $kg^{-1}$ (1 $\mu l$ $g^{-1}$) was administered intraperitoneally daily from postnatal days (P)1 to P9 using a 30-G needle (Hamilton syringe). Control animals received an equal volume of PBS. The pups were returned to their dams after the injection. The total volume injected depended on the individual animal's weight. Mice were sacrificed using a lethal intraperitoneal injection of ketamine/xylazine, and eyes were enucleated at P10. Body weight was recorded, and blood glucose levels were determined using (glucose strip). Five mouse pups were injected intravitreally with 0.5 $\mu l$ STZ (2 $mg$ $ml^{-1}$) at P1 to assess the chemical toxicity of STZ on the developing retina. 0.5 $\mu l$ PBS was injected at the contralateral eye of the same pup as control. Recombinant mouse APN (a combination of high-molecular-weight, hexamer and trimer APN, i.p. 0.6 $\mu g$ $g^{-1}$ in PBS, ALX-522-059-C050, Enzo) or AdipoRon (oral gavage 50 $mg$ $kg^{-1}$ in corn oil, Cayman) was administered from P7 to P9. Both male and female mice were used. Littermates were administrated vehicle as control. Retinal vasculature was examined at P10.

## Quantification of retinal vasculature

APN knockout ($Apn^{-/-}$) mice (JAX, #008195) were used. $Apn^{-/-}$ mice have been backcrossed to C57BL/6J for 11 generations. P10 $Apn^{-/-}$ and C57BL/6J (used as controls recommended by the vendor) retinas were dissected, permeabilized with 1% Triton X-100–PBS, and stained overnight at room temperature with fluorescent Griffonia Bandeiraea Simplicifolia Isolectin B4 (Alexa Fluor 594, I21413, Molecular Probes, 10 $\mu g$ $ml^{-1}$) in 1 mM $CaCl_2$ in PBS. For quantification of retinal network formation, 4–5 images between the optic nerve head and the leading edge of vessels in the superficial or deep vascular layers were taken at 200× magnification on a Zeiss AxioObserver.Z1 microscope. The images were analyzed using "Angiogensis analyzer" plugin in ImageJ. Background spurious staining was cleaned manually and the image was converted to 8-bit before analysis. Parameters included were the number of meshes and total vessel length. For the quantification of deep retinal vascular coverage, the image of each quadrant was taken at 200× magnification and merged to form one image with AxioVision 4.6.3.0 software. The images were saved in grayscale. The optic nerve head was used as the reference point for the four quadrants. The vascular area and total retinal area were quantified for each quadrant and summed for the entire retina. The percentage of deep vascular coverage over the total retinal area was calculated.

## Electroretinography

Visual function was assessed with electroretinography (ERG) for P30 mice as previously described (Sapieha *et al*, 2012). Mice were dark-adapted overnight and anesthetized (ketamine/xylazine), and pupils were dilated (Cyclomydril; Alcon, Fort Worth, TX, USA). The stimuli were "green" light-emitting diode flashes of doubling intensity from ~0.0064 to ~2.05 cd s $m^{-2}$ and then "white" xenon-arc flashes from ~8.2 to ~1050 cd s $m^{-2}$. The stimuli were delivered using a Colordome Ganzfeld stimulator (Diagnosys LLC, Lowell, MA). The *a*-waves representing saturating photoresponses were estimated by fitting the free parameters in a model of the biochemical processes involved in the activation of phototransduction to the electroretinographic *a*-waves (Lamb & Pugh, 1992; Pugh & Lamb, 1993; Hood & Birch, 1994). The saturating responses of *b*-waves were derived from the Naka–Rushton equation (Fulton & Rushton, 1978). The oscillatory potentials (OPs) characterize the inner retina cell activity, which is distinct from those generate *a*-waves and *b*-waves (Sapieha *et al*, 2012). All electroretinographic data were presented as the log change from normal ($\Delta$LogNormal). To investigate the long-term impact of APN on retinal function, 25 $mg$ $kg^{-1}$ STZ was intraperitoneally injected from P2 to P12, and msAPN or vehicle was given from P7 to P39. ERG was conducted at P40.

## Quantification of serum total APN, triglycerides, and insulin levels

Serum APN levels were assayed using a human APN ELISA kit (E091M, Mediagnost). Serum triglycerides levels were measured using the Wako L-Type TG M test. Serum insulin was assayed using mouse ultrasensitive insulin ELISA kit (80-INSMSU-E01, Alpco).

## Laser capture microdissection

P10 eyes were enucleated and embedded in OCT compound. The eyes were sectioned at 12 $\mu m$ in a cryostat, mounted on RNase-free polyethylene naphthalate glass slides (11505189, Leica), and immediately stored at $-80°C$. Slides containing frozen sections were fixed in 70% ethanol for 15 s, followed by 30 s in 80% ethanol and 30 s in absolute ethanol, and then washed with DEPC-treated water for 15 s. Sections were stained with fluoresceinated Isolectin B4 (Alexa Fluor 594—I21413, Molecular Probes, 1:50 dilution in 1 mM $CaCl_2$ in PBS) and treated with RNase inhibitor (03 335 399 001, Roche) at room temperature for 3 min. Laser dissection of retinal blood vessels and retinal neuronal layers was performed immediately thereafter with the Leica LMD 6000 system (Leica Microsystems), and samples were collected directly into lysis buffer from the RNeasy Micro kit (Qiagen, Chatsworth, CA, USA).

## Examination of hyaloid vessel persistence

P8 eyes were enucleated and fixed in 4% paraformaldehyde for 1 h at room temperature. The eyes were then intravitreously injected with 1.5% agarose in PBS (pre-warmed at 37°C). After 20 min on ice, the cornea was removed, followed by the iris, sclera, choroid, and RPE. PBS was injected between the retina and the vitreous to separate the two structures. The retina was then gently peeled off from the vitreous body. The lens was then removed and the hyaloid vessels were visualized with 4′,6-diamidino-2-phenylindole (DAPI, Vector, Burlingame, CA, USA). The images were taken at 50× magnification on a Zeiss AxioObserver.Z1 microscope. The total

number of persistent hyaloid vessels branching from the hyaloid artery was manually counted.

Hyaloid persistence at P30 was examined with fluorescein angiography. Mice were anaesthetized with ketamine/xylazine, the pupil was dilated with Cyclomydril (Alcon), and the cornea was anesthetized with proparacaine; 10 mg ml$^{-1}$ fluorescein AK-FLUOR$^®$ (NDC 17478-101-12, Akorn) was intraperitoneally injected at 10 μl g$^{-1}$ body weight. Images were taken with the Micro IV retinal imaging system.

## Quantitative proteomics of retinal metabolic enzymes

### Sample preparation

Both retinas from each mouse were pooled and homogenized in 200 μl Holt's lysis buffer. For each homogenate, the protein concentrations were measured with by the Pierce™ BCA protein assay, and the volume of homogenate containing 75 μg total protein was taken for analysis. Sodium dodecyl sulfate (SDS) was added to a final concentration of 1%. Bovine serum albumin, 8 pmol, was added as a non-endogenous internal standard. The samples were heated at 70°C for 15 min to equilibrate the standard before adding 1 ml acetone and incubating overnight at −20°C to precipitate the proteins. The precipitated proteins were pelleted by centrifugation at 10,000 × g for 10 min, and the acetone was poured off and any residual acetone removed in a SpeedVac. The dried protein pellet was reconstituted in 75 μl Laemmli sample buffer; 20 μl aliquots, containing 20 μg total protein, were run 1.5 cm into a 12.5% SDS–PAGE gel (Bio-Rad). These short-run gels were fixed, washed, and stained with Pierce GelCode Blue. Each 1.5-cm lane was cut from the gel as a single sample, divided into smaller pieces, and washed in ethanol/water/acetic acid (45/45/10) to remove the stain.

### In-gel digestion

The proteins contained in the samples were reduced with DTT, alkylated with iodoacetamide, and digested with 1 μg trypsin for overnight at room temperature. The peptides were extracted in methanol/water/formic acid (45/45/10) and the extract evaporated to dryness in a SpeedVac. The dried peptide mixture was reconstituted in 150 μl 1% acetic acid for analysis.

LC-tandem MS analysis using selected reaction monitoring (SRM) (Kinter & Kinter, 2013). The mass spectrometry system was a Thermo Scientific TSQ Vantage with an Eksigent splitless nanoflow HPLC system. The samples (7 μl aliquots) were injected onto a 10 cm × 75 μm i.d. capillary column packed with Phenomenex Jupiter C18 reversed-phase beads. The column was eluted at 150 nl min$^{-1}$ with a 60-min linear gradient of acetonitrile in 0.1% formic acid. The SRM assays were developed to monitor two peptides per protein. Each peptide was monitored in a 6-min window centered on the known elution time of the peptide (Kinter & Kinter, 2013). The data were processed using the program Skyline to determine the respective chromatographic peak areas (MacLean et al, 2010). The response for each protein was taken as the geometric mean of the two peptides monitored. Changes in the relative abundance of the proteins were determined by normalization to the BSA internal standard. The data was presented as heat maps and the raw data are shown in Table EV1.

## Photoreceptor (661W) cell culture

Cone photoreceptor 661W was originally cloned from retinal tumors of a transgenic mouse line that expresses the simian virus (SV) 40 T antigen under control of the human interphotoreceptor retinol-binding protein (IRBP) promoter and characterized by immunohistochemistry and immunoblot (Tan et al, 2004). Cells were cultured at 37°C, 5% CO$_2$ in a humidified atmosphere in Dulbecco's modified Eagle's medium (DMEM, #1196502, Gibco) supplemented with 10% fetal bovine serum (#S12450, Atlanta Biologicals) and 1% penicillin/streptomycin. No mycoplasma contamination of the cells was detected using ATCC universal mycoplasma detection kit (30–1,012 K). An equal number of cells per well was plated on the 6-well dish and cultured to 80 to 90% confluence. The cells were treated with 50 μM AdipoRon or DMSO as vehicle control for 24 h and collected for protein and RNA.

## Oxygen consumption rate and extracellular acidification rate

Oxygen consumption rate (OCR) and extracellular acidification rate (ECAR) were measured using a Seahorse XF$^e$96 Flux Analyzer (Joyal et al, 2016). For retinal OCR and ECAR ex vivo, whole retinas were isolated and 1-mm punches were loaded into wells of a 96-well spheroid plate. Retinal punches were incubated in assay medium [(DMEM, D5030, Sigma) supplemented with 25 mM glucose, 5 mM HEPES and 2 mM glutamine, pH 7.4] with the presence of 0.1 pg ml$^{-1}$ msAPN or PBS (vehicle control) for 2 h before taking measurements. For 661W cells in vitro, 4,000 cells per well were placed in each well of a 96-well plate for 48 h and treated with 0.05–0.5 μM AdipoRon or DMSO (vehicle control) for 6 h before taking measurements. To determine the amount of proton uncoupling (OCR independent of adenosine triphosphate production), 1.5 μM oligomycin, an adenosine triphosphate synthase inhibitor was used for 661W cells. Due to the reported impact of oligomycin on retinal OCR ex vivo (Kooragayala et al, 2015), we omitted this step for retinal punches. To determine the maximal mitochondrial respiration, the uncoupling agent carbonyl cyanide-4-trifluoro-methoxy phenylhydrazone (FCCP, 0.5 μM for retinal punches; 1 μM for 661W cells) was utilized. The non-mitochondrial respiration rate was measured with the addition of rotentone (complex I inhibitor) and antimycin A (complex III inhibitor) (2 μM for retinal punches; 1 μM for 661W cells).

## Examination of mitochondrial morphology

661W cells were placed on a glass coverslip in a 6-well plate (40,000 cells per well) at day (D) 0. At D1, the cells were treated with 50 mM glucose with the presence of 0.5 nM AdipoRon or DMSO (vehicle control) for 24 h. 50 mM mannitol was used as a control for osmotic stress. At D2, the cells were incubated with pre-warmed DMEM supplemented with 100 nM MitoTracker (#M7512, Invitrogen) for 30 min at 37°C. After washing with PBS, the cells were fixed with pre-warmed 4% paraformaldehyde for 15 min at 37°C and then incubated with 0.2% Triton X-100 PBS for 10 min at room temperature. The cells were blocked with 1% bovine serum albumin and stained with 1:40 F-actin (#A12379, Life Technologies) for cell skeleton and DAPI for cell nuclei. The images were taken at 400× magnification on a Zeiss AxioObserver.Z1 microscope. The

mitochondrial morphology was categorized as tubular, intermediate, and fragmented (Eura *et al*, 2006; Zhao *et al*, 2009). The number of cells was counted, and the percentage of cells within each mitochondria category versus the total number of cells per imaging field was calculated.

## Real-time PCR

P10 retinas or 661W cells were lysed with QIAzol lysis reagent and incubated on ice for 15 min; 20% chloroform was added and incubated for 5 min at room temperature. RNA was extracted according to the manufacturer's instructions using a PureLink® RNA Mini Kit (#12183018A, Ambion). RNA was then reverse-transcribed using iScript™ cDNA synthesis kit (#1708891, Bio-Rad). qPCR were performed for *Apn:* 5′-GAAGCCGCTTATGTGTATCGC-3′, 5′-GAATG GGTACATTGGGAACAGT-3′; *AdipoR1:* 5′-TCTTCGGGATGTTCTTCC TGG-3′, 5′-TTTGGAAAAAGTCCGAGAGACC-3′; *AdipoR2:* 5′-GGAGT GTTCGTGGGCTTAGG-3′, 5′-GCAGCTCCGGTGATATAGAGG-3′; *Cs:* 5′-CCAGTTCACCTCCCCCTCCTGG-3′, 5′-GGGCCTGTCCCTGGCGTA GA-3′; *Fis1:* 5′-AGAGCACGCAATTTGAATATGCC-3′, 5′-ATAGTCCC GCTGTTCCTCTTT-3′; *Hk1:* 5′-CAAGAAATTACCCGTGGGATTCA-3′, 5′-CAATGTTAGCGTCATAGTCCCC-3′; *Mff:* 5′-ATGCCAGTGTGATA ATGCAAGT-3′, 5′-CTCGGCTCTCTTCGCTTTG-3′; *Mfn1:* 5′-ATGGC AGAAACGGTATCTCCA-3′, 5′-CTCGGATGCTATTCGATCAAGTT-3′; *Mfn2:* 5′-AGAACTGGACCCGGTTACCA-3′, 5′-CACTTCGCTGATAC CCCTGA-3′; *Pdgfb:* 5′-TGCTACCTGCGTCTGGT-3′, 5′-GATGAGC TTTCCAACTCGACTC-3′. Quantitative analysis of gene expression was generated using an Applied Biosystems 7300 Sequence Detection System with the SYBR Green Master mix kit, and gene expression was calculated relative to *Cyclophilin A* (retinas) or *β-actin* (661W cells) using the $\Delta\Delta C_t$ method.

## Western blot

40 μg protein lysate from 661W cells were used to detect the levels of phosphor-AMPK (Ma *et al*, 2016) (p-AMPK, 1:1,000, #2535, Cell Signaling) and AMPK (Almudena *et al*, 2016) (1:1,000, #2532, Cell Signaling) overnight at 4°C. Signals were detected using 1:5,000 corresponding horseradish peroxidase-conjugated secondary antibodies and enhanced chemiluminescence (ECL, Pierce).

## Preparation of AAV2-RK-sh*Pdgfb*, AAV2-RK-sh*Vegfa* vector, and AAV2 virus

ShRNA against mouse *Pdgfb* were designed as previously published (Paddison *et al*, 2004). The template oligonucleotides contain miR-30 microRNA, miR-30 loop and shRNA including the sense and the antisense were synthesized (Invitrogen). The target sequence was as follows: *Pdgfb:* 5′-GGATCCCATTCCTGAGGAACT-3′; Sh*Vegfa*: 5′-AACCTCACCAAAGCCAGCACAT-3′; and *Vegfa*: 5′-AACCTCAC-CAAAGCCAGCACAT-3′. DNA fragments were inserted into human rhodopsin kinase (RK)-GFP-miR30 vector (Joyal *et al*, 2016). The *Pdgfb* knockdown efficiency was tested in 661W cells. Recombinant AAV2 vectors were then produced as before (Joyal *et al*, 2016). AAV2 vector, rep/cap packaging plasmid, and adenoviral helper plasmid were mixed with polyethylenimine in DMEM and transfected into HEK293T cells (CRL-11268, ATCC). After 56 h, cells were collected for virus purification using iodixanol. This protocol

---

## The paper explained

### Problem
Retinopathy of prematurity is a common complication in premature infants and a leading cause of blindness in children. The factors and neural cells determining normal vascular growth are not well defined even though delayed vascular growth (phase I) drives vision-threatening neovessel growth (phase II). Hyperglycemia and low adiponectin (APN, a key metabolic regulator) levels correlate with the progression of phase II ROP. We examined whether these effects are driven primarily by dysregulated photoreceptor metabolism which delays normal retinal vascularization at phase I ROP.

### Results
The present study describes that in premature infants, low APN levels correlate with hyperglycemia and delayed retinal vascular formation. Experimentally in a newly established neonatal mouse model of hyperglycemic ROP, hyperglycemia causes photoreceptor dysfunction and delays neurovascular maturation. Activation of the APN pathway normalizes vascular growth and restored retinal function. APN deficiency decreases retinal metabolism particularly in photoreceptors and decreases photoreceptor-derived growth factors for vascular development.

### Impact
Our work reveals the impact of hyperglycemia and APN on normal retinal vessel growth in phase I ROP and explores a new therapeutic intervention (stimulation of the APN pathway) at early stages to improve hyperglycemia-associated retinal neurovascular disorders like retinopathy of prematurity in preterm infants with implications for diabetic retinopathy in adults.

---

was also used to prepare a control AAV2-shControl (GFP). 0.5 μl virus (about $2 \times 10^{12}$ gc ml$^{-1}$) was injected into the subretinal space in WT mouse retina at P1. The samples were collected at P10 for the examination of retinal vascular development.

## Statistical methods

Clinical data are presented as mean ± SEM. For continuous variables, the *t*-test was used to test differences between groups. *P*-values < 0.05 were considered significant. Statistical analyses were performed using the program package IBM SPSS statistics 23 for Microsoft Windows (IBM, Armonk, NY, USA). Animal data are presented as mean ± SEM. All data were used except for low-quality images that were not sufficient for analysis. The number per group (*n*) was shown in figure legends. For drug treatments, the mice were randomly assigned to treatment and vehicle control in the same litter. For *in vitro* study, the experiment was repeated at least three independent times. The experimenters were blinded to the treatment conditions. Sample size was computed by G*Power3 (Faul *et al*, 2007) to allow 80% power (set β to 0.2) and set α to 0.05. D'Agostino & Pearson omnibus normality test was used to confirm normal distribution. F-test was used to compare the variance. Two-tailed unpaired *t*-test or ANOVA with Bonferroni's multiple comparison test was used for comparison of results as specified (Prism v5.0; GraphPad Software, Inc., San Diego, CA, USA). Data with non-Gaussian distribution were analyzed by Kruskal–Wallis test followed by Dunn's multiple comparisons test (nonparametric, three groups) or Mann–Whitney

test (nonparametric, two groups). Statistically significant difference was set at $P \leq 0.05$. The results of statistical analysis are shown in Table EV2.

## Study approval

The clinical study was approved by the Regional Ethical Review Board in Lund, Sweden. Inclusion criteria were a gestational age < 31 weeks at birth, absence of major congenital anomalies, and written informed parental consent from both parents. All pregnancies were dated by ultrasound at 17–18 gestational weeks. Fifty-one infants completed the study until term age and longitudinal plasma glucose and adiponectin concentrations were analyzed in all infants. Written informed consent was obtained from parents for all study infants. The prospective longitudinal cohort study was conducted between January 2005 and May 2007. Previous data reported on this cohort include the relationship between circulatory insulin-like growth factor concentrations and postnatal growth, brain volumes, and outcome (Hellstrom *et al*, 2003; Hansen-Pupp *et al*, 2011, 2013; Fu *et al*, 2015). All animal studies adhered to the Association for Research in Vision and Ophthalmology Statement for the Use of Animals in Ophthalmic and Vision Research and were approved by the Institutional Animal Care and Use Committee at Boston Children's Hospital.

## Data availability

The raw data of retinal proteomics was published in BioStudies (https://zenodo.org/, accession number 1054112).

**Expanded View** for this article is available online.

## Acknowledgements

We thank Enzo Inc. for providing recombinant mouse adiponectin. We thank Dr. Jing Chen for her expert advice. We also thank Mari Gantner, Ricky Z. Cui, Christian G. Hurst, Nicholas J. Saba, Thomas W. Fredrick, Peyton C. Morss, and Elizabeth P. Moran for their technical support. The work is supported by NIH EY024864, EY017017, BCH IDDRC (1U54HD090255), Lowy Medical Research Institute, European Commission FP7 project 305485 PREVENT-ROP (LEHS); The Swedish Research Council (DNR# 2011-2432) and Gothenburg County Council (ALFGBG-426531) long-term support by De Blindas Vänner and Kronprinsessan Margaretas Arbetsnämnd för synskadade (AH); project 305485 PREVENT-ROP (VINNOVA 2009-01152) from the European Commission FP7 (CAL); Swedish Research Council 2014-3140, governmental ALF research grants to Lund University and Lund University Hospital, European Commission (FP7, Project 305485 PREVENT-ROP) (DL); NIH P30AG050911 and the Harold Hamm Diabetes Center at the University of Oklahoma (MK). ZF is supported by Knights Templar Eye Foundation (#76293) and Blind Children's Center (#89282); Deutsche Forschungsgemeinschaft (DFG), Li2650/1-1 (RL); Knights Templar Eye Foundation (CHL); Boston Children's Hospital OFD/BTREC/CTREC Faculty Career Development Grant (YS).

## Author contributions

Designing research studies: ZF, LEHS, CAL, IHP, DL, ST, and AH; Acquiring and analyzing data: ZF, LEHS, CAL, AH, IHP, DL, JDA, and MK; Writing original draft: ZF; Writing review and editing: LEHS, AH, CAL, IHP, and ST; Funding acquisition: LEHS and AH; Conducting experiments: ZF, CAL, RL, ZW, YS, YG, SSM, SBB, IA, C-HL, JDA, MK, MTC, RD, AP, and SSC; Supervision, ST.

## Conflict of interest

The authors declare that they have no conflict of interest.

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
