## [Review Process File · EMBO Molecular Medicine]

Photoreceptor glucose metabolism determines normal retinal vascular growth

Zhongjie Fu, Chatarina A. Löfqvist, Raffael Liegl, Zhongxiao Wang, Ye Sun, Yan Gong, Chi-Hsiu Liu, Steven S. Meng, Samuel B. Burnim, Ivana Arellano, My T. Chouinard, Alexander Poblete, Steven S. Cho, James D. Akula, Michael Kinter, David Ley, Ingrid Hansen Pupp, Saswata Talukdar, Ann Hellström, and Lois E. H. Smith

Corresponding author: Lois Smith, Department of Ophthalmology, Boston Children's Hospital

Review timeline:

Submission date:	01 May 2017
Editorial Decision:	06 June 2017
Revision received:	28 September 2017
Editorial Decision:	12 October 2017
Revision received:	25 October 2017
Accepted:	30 October 2017

Transaction Report:

Editor: Céline Carret

1st Editorial Decision

06 June 2017

Thank you for the submission of your manuscript to EMBO Molecular Medicine. We have now heard back from the two referees whom we asked to evaluate your manuscript.

You will see that both reviewers find the study interesting and overall well executed. Nevertheless, both referees request additional explanations and clarifications and suggest further analyses to make the study even more compelling. Please make sure to use the right statistical tests throughout the manuscript, i.e. t-test should not be used for multiple comparison. Should you be able to address these criticisms in full, we would be willing to consider a revised manuscript.

I look forward to receiving your revised manuscript.

***** Reviewer's comments *****

Referee #2 (Remarks):

Fu and colleagues report an interesting study focusing on adiponectin's function in hyperglycemia related "retinopathy of prematurity" (ROP). They report that low adiponectin correlated with delayed retinal vascularization in premature infants. They also found adiponectin deficiency delays hyaloid regression and retinal vascularization in neonatal mice. Activation of the APN pathway by

recombinant adiponectin or the agonist AdipoRon prevents retinopathy in an STZ-induced hyperglycemic mouse model. They further explore the mechanism connecting adiponectin /AdipoR to AMPK activation and mitochondrial function, which improves glucose metabolism in photoreceptor cells. Finally, they identified PDGFB as a responder effector of adiponectin function, mediating the generation of the vascular network. A timely and well performed set of studies. A few minor concerns:

1. Serum APN is generally reduced in established obesity/diabetes. In the STZ hyperglycemic retinopathy neonatal mice, both circulating adiponectin levels and AdipoR1 (the dominant receptor) expression in retinas are paradoxically higher, suggesting that the adiponectin pathway may be preserved and in fact enhanced under these conditions. What is the possible mechanism for this phenomenon in these STZ neonatal mice? Hyperglycemia is driving the retinopathy during the adiponectin /agonist treatment, is hyperglycemia corrected under these conditions?
2. The retinas are metabolically very active, as there is almost no reserve capacity in the vehicle-retinas (Fig.4B). In vivo treatment with the agonist AdipoRon clearly increased maximal oxygen consumption. Are these measurements done in the retinas from hyperglycemic or euglycemic mice? Does oxygen consumption decrease in hyperglycemic retinas, and is that improved by adiponectin?
3. In 661W cells, AMPK is activated by AdipoRon treatment. Is the retinal AMPK activation required for adiponectin's effect in glucose metabolism and prevention of retinopathy?
4. Pdgfb is known to promote retinal vascular growth, as also shown in Fig 4, knockdown pdgfb in photoreceptor cells impairs the formation of the retinal vascular network. Does an intervention in the adiponectin receptor pathway reverse these defects?
5. Can the mice that have undergone retinopathy, upon activation of the adiponectin pathway, eventually develop normal eye function when grown up?
6. Is this strictly a glucotoxic phenomenon or is there a lipotoxic component involved as well? With STZ treatment, there is obviously a high degree of lipolysis ongoing, presumably also in these very young mice. What are the lipid levels under these conditions, and could this potentially involve the ceramide pathway?

Referee #3 (Remarks):

This is a very interesting and potentially important study that combines clinical and experimental investigations to elucidate a new mechanism for the development of retinopathy of prematurity (ROP), a sight-threatening disease in premature infants. Clinically, reduced levels of adiponectin (Apn) were found to be correlated with hyperglycemia and delayed retinal vessel formation in premature infants. Experimental studies using a newly developed mouse model of hyperglycemia-associated retinopathy and 661w cell line demonstrate that depletion of Apn or activation of Apn pathway affects hyaloid vessel regression and retinal vascular formation in normal or hyperglycemic mice. Further experiments show that Apn increases retinal and photoreceptor metabolism and stimulates photoreceptor derived Pdgfb production. These findings clearly suggest a role of Apn in regulation of photoreceptor metabolism and vascular formation and therefore may serve as a potential target for ROP.

Major points:

1. A major contribution of this manuscript would be the generation of a new model for study ROP-related hyperglycemia-induced retinopathy. However, some important information appears to be missing. For example, what is the timing for the development of hyperglycemia and decrease of insulin production in this model. How about the survival rate of neonatal mice receiving STZ injection? In addition, injection of STZ from P1 - P9 only affects the deep layer but not superficial layer vascular formation. Can this phenomenon be explained by the timing for the development of hyperglycemia?
2. It is hypothesized that hyperglycemia suppresses photoreceptor metabolism resulting in delayed retinal vascularization. However, direct evidence for this is lacking (Apn has no effect on hyperglycemia). Perhaps it would be more convincing to demonstrate the role of hyperglycemia in inducing photoreceptor dysfunction by supplementation of insulin to the hyperglycemic neonatal mice.
3. While most data were derived from a large number of animals and appear to be very compelling, the quality of the deep layer vasculature seems to be suboptimal. Further, some representative images do not show the changes as indicated in the graph. For example, Figure 3C: the images on the left show fewer hyaloid vessels in Apn^{-/-} retinas, while the graph shows an increase in these

retinas.

4. Is AMPK activation altered by AdipoRon injection and in Apn^{-/-} retinas in vivo? These data would be helpful in supporting the hypothesis (Figure 5).
5. It is less clear how Apn regulates VEGFA expression, although Figure 5 indicates with Apn photoreceptors produce less VEGF (+PDGFB, -VEGF)? There is no data provided in this regard.
6. The use of "photoreceptors" instead of "661w cells" is confusing and misleading, since changes in photoreceptors generally refer to results from in vivo studies. Furthermore, 661w cells is a cone photoreceptor cell line. The results derived from the in vitro study should be carefully interpreted

Minor points:

7. The definition of HAR should be consist in the manuscript. In the legends for Figure 2: hyperglycemia-associated retinopathy (HAR); Figure3: ...hyperglycemia-associated retinal vascularization (HAR) delay.... Which is correct?
8. Legend for Figure 4: Please specify the age of the mice used in each assay. (B): is the difference significant? (C): please clarify "In Apn^{-/-} versus WT STZ" - were these mice hyperglycemic?
9. Figure EV3b, c shows that normal or hyperglycemic Apn^{-/-} retinas display significant reduction in superficial layer vascular formation. This information is missing in the results section. The significance and mechanism for this defect should discussed. In contrast, there is only a modest decrease in the deep layer vascular formation in Apn^{-/-} retinas (Figure 3E). Is there a possible explanation for this?
10. Symbols for indications of significance are missing in some figures.

1st Revision - authors' response

28 September 2017

Referee #2 (Remarks):

Fu and colleagues report an interesting study focusing on adiponectin's function in hyperglycemia related "retinopathy of prematurity" (ROP). They report that low adiponectin correlated with delayed retinal vascularization in premature infants. They also found adiponectin deficiency delays hyaloid regression and retinal vascularization in neonatal mice. Activation of the APN pathway by recombinant adiponectin or the agonist AdipoRon prevents retinopathy in an STZ-induced hyperglycemic mouse model. They further explore the mechanism connecting adiponectin /AdipoR to AMPK activation and mitochondrial function, which improves glucose metabolism in photoreceptor cells. Finally, they identified PDGFB as a responder effector of adiponectin function, mediating the generation of the vascular network. A timely and well performed set of studies. A few minor concerns:

1. Serum APN is generally reduced in established obesity/diabetes. In the STZ hyperglycemic retinopathy neonatal mice, both circulating adiponectin levels and AdipoR1 (the dominant receptor) expression in retinas are paradoxically higher, suggesting that the adiponectin pathway may be preserved and in fact enhanced under these conditions. What is the possible mechanism for this phenomenon in these STZ neonatal mice? Hyperglycemia is driving the retinopathy during the adiponectin /agonist treatment, is hyperglycemia corrected under these conditions?

Responses: We thank the reviewer for the comments on the merit of our study and we understand the reviewer's concern.

In STZ-induced mouse neonates, serum insulin levels were decreased and the APN pathway was enhanced. Pharmaceutical activation of the APN pathway promoted and deficiency of APN further inhibited the deep retinal vascularization, suggesting that activation of the APN pathway may serves as a compensatory response with insulin deficiency. The phenomenon seen here is actually not unique. In type 1 diabetes (T1D), circulating APN levels are higher than non-diabetic controls (Pereira et al, 2012). Serum APN levels in T1D positively correlate with increased insulin sensitivity, plasma total antioxidant status, as well as with less renal disease (Pereira et al, 2012; Prior et al, 2011; Yuan et al, 2014). APN may be strongly related to insulin resistance with longer duration of T1D as a strong negative relationship between APN and log insulin dose is increased

(LeCaire & Palta, 2015). It has been found that APN mediated by APPL1 to promote insulin sensitization in skeletal muscles (Deepa & Dong, 2009).

However, in obesity, fat accumulation causes dysregulation of adipokine, which in turn is strongly correlated with obesity-related disorders. In type 2 diabetes (T2D), circulating APN levels are decreased versus healthy controls (Yilmaz et al, 2004). In T2D, serum APN levels are inversely related to insulin resistance and patients with abdominal obesity have lower APN levels (Aleidi et al, 2015). Serum APN levels are inversely associated with the incidence of T2D (Lindsay et al, 2002; Yamamoto et al, 2014). In addition, there is also a decline in APN receptors AdipoR1 and AdipoR2 mRNA expression in obesity/T2D, further attenuating the APN effects (Kadowaki et al, 2007).

Therefore, we speculate that the APN pathway is activated to protect against the cell metabolic stress with lack of insulin. However, under obese/T2D conditions, the production of APN and its receptors are affected and thus the modulatory effects of APN on metabolism are dampened. In neonatal STZ, although we did not observe a significant impact of APN on circulating blood glucose levels (mean blood glucose at P10 *Apn*^{-/-} HAR mice: 206±8mg/dl; WT HAR mice: 197±6mg/dl), we did observe further increased circulating triglyceride levels with APN deficiency (as shown below). In addition, we also found that APN deficiency decreased the local production of retinal metabolic enzymes involved in glucose pathway and mitochondrial activity (**Figure 4A**), as well as a direct impact of APN treatment on retinal mitochondrial respiration (**Figure 4B**). Taken together, we summarized that activation of the APN pathway compensated for the reduced circulating insulin levels, which in turn promoted retinal metabolism and neurovascular development in the HAR model.

2. The retinas are metabolically very active, as there is almost no reserve capacity in the vehicle-retinas (Fig.4B). In vivo treatment with the agonist AdipoRon clearly increased maximal oxygen consumption. Are these measurements done in the retinas from hyperglycemic or neuglycemic mice? Does oxygen consumption decrease in hyperglycemic retinas, and is that improved by adiponectin?

Responses: We appreciate the reviewer's valuable comments. The measurement of AdipoRon on retinal oxygen consumption rate (OCR) was originally obtained from normal WT retinas when we tried to prove the concept that activation of the APN pathway increased retinal OCR.

We did observe an overall reduction in OCR in hyperglycemic versus normal control retinas (**Figure EV3, A**). The mild changes seen possibly resulted from the fact that non-littermate controls were used. Unfortunately, we noticed that if we used mice in the same litter for controls, the STZ-induced mouse pups were weaker than the vehicle (PBS)-treated pups, which led to less capability to fight for the nursing dam's milk and in turn they grew more slowly than the control pups. Therefore, in the HAR model, we treated all mice from same litter with either STZ or PBS for the phenotypic studies. The number of mouse pups was adjusted in PBS-treated group to achieve a similar range of body weights as mouse body weight may affect the development of retinopathy during developmental stage (Connor et al, 2009).

In the Seahorse analysis of retinal punches *ex vivo*, we noticed that oligomycin treatment had a negative impact on the induction of maximal respiration later, as previously reported (Kooragayala et al, 2015). Therefore, in our later studies, we did not treat retinas with oligomycin in order to achieve maximal respiration. We speculate that the lack of reserve metabolic capacity in normal WT retinas might be due to the vehicle (DMSO, the diluting solution for AdipoRon) used. Therefore, we repeated the experiment with either msAPN or vehicle (PBS) treatment in WT HAR mice. Vehicle-treated retinas showed increased respiration after FCCP and msAPN significantly increased retinal OCR (**Fig. 4B**).

3. In 661W cells, AMPK is activated by AdipoRon treatment. Is the retinal AMPK activation required for adiponectin's effect in glucose metabolism and prevention of retinopathy?

Responses: To examine if activation of AMPK is essential for APN to exert a protective effect on hyperglycemic retinopathy, we treated the STZ-induced mice with mouse recombinant APN (msAPN) and Compound C (AMPK inhibitor (Liu et al, 2014; McCullough et al, 2005)) or vehicle from P7-9. At P10, we found that co-treatment with Compound C significantly reduced deep retinal vascular formation versus vehicle control in the presence of msAPN administration (**Figure EV3, H**). Therefore, AMPK activation was required for APN protection against hyperglycemia-induced retinal vascular changes.

Figure EV3: (H) Deep retinal vascular formation in WT HAR mice co-treated with msAPN (0.6µg/g) and Compound C (AMPK inhibitor, 2µg/g) or vehicle (DMSO, the diluting solution for Compound C) from P7 to P9. Unpaired t test.

4. Pdgfb is known to promote retinal vascular growth, as also shown in Fig 4, knockdown pdgfb in photoreceptor cells impairs the formation of the retinal vascular network. Does an intervention in the adiponectin receptor pathway reverse these defects?

Responses: In our drug intervention study, we found that msAPN administration showed better retinal vascular formation than vehicle control (**Fig. 3F**). We now further measured local gene expression of *pdgfb* in retinas after msAPN administration. msAPN increased retinal *pdgfb* expression (**Fig. 4H**). APN deficiency led to reduced *pdgfb* expression, particularly in photoreceptors (**Fig. 4I**). Therefore, in summary, interventions affecting the APN pathway could increase retinal expression of *pdgfb*, which in turn could promote retinal vascular growth.

Fig. 4: (H) msAPN treatment increased retinal expression of *Pdgfb* in WT HAR mice. Littermate controls were used. (I) *Pdgfb* expression in photoreceptors (ONL from LCM) from $Apn^{-/-}$ versus WT STZ mice (LCM and qRT-PCR). t test.

5. Can the mice that have undergone retinopathy, upon activation of the adiponectin pathway, eventually develop normal eye function when grown up?

Responses: Yes, msAPN treated mice (0.6ug/g) versus vehicle control (PBS) for up to one month in the HAR model had significantly better retinal function detected by electroretinography (ERG). In the HAR model, both rod (**Figure 5A**) and cone (**Figure 5E**) signals detected by ERG were significantly decreased. msAPN treatment significantly increased post-receptor b-wave amplitude; there was also a non-significant trend of increase in photoreceptor a-wave sensitivity and amplitude (**Figure 5D**). As post-receptor responses positively correlated with the sum of changes in photoreceptors (**Figure 5B**), our findings indicated that msAPN might rescue rod photoreceptor function in HAR. msAPN treated mice also demonstrated significantly better cone function (reflected by post-receptor response V_m) (**Figure 5E**). Therefore, our findings suggested that activation of the APN pathway can improve eye function long term in the HAR model.

Figure 5: Hyperglycemia was induced by 25mg/kg STZ (i.p., daily from P2-12). The mouse pups received msAPN (0.6ug/g) or vehicle (PBS) injection daily from P7-39. At P40, rod (**D**) and cone (**E**) ERG were conducted. Rod response: amplitude ($R_{m_{p3}}$) sensitivity (S); post-receptor response: amplitudes (V_m), sensitivity (σ) as well as total retinal sensitivity (S_m). $n=7$ to 12 per group. ** $P < 0.01$, ANOVA (D); t test (E).

6. Is this strictly a glucotoxic phenomenon or is there a lipotoxic component involved as well? With STZ treatment, there is obviously a high degree of lipolysis ongoing, presumably also in these very young mice. What are the lipid levels under these conditions, and could this potentially involve the ceramide pathway?

Responses: Thanks for raising this important question and we agree with the reviewer that both glucose and lipid metabolic alterations might contribute to the disease progression.

In the HAR model, serum triglyceride levels were highly increased after the induction of hyperglycemia with STZ treatment (**Figure 2D, Figure EV1, C**). Our preliminary data also showed that retinal ceramide levels were higher in APN-deficient mice versus WT mice in HAR model (unpublished data). Ceramide induces photoreceptor apoptosis (German et al, 2006). A strong correlation between ceramide signaling and homeostasis of photoreceptor and retinal pigment epithelial cells is also suggested in retinal degeneration (Chen et al, 2012; Strettoi et al, 2010). APN receptor adipoR1 and adipoR2 possess ceramidase activity to convert ceramide to anti-apoptotic metabolite sphingosine-1-phosphate (Holland et al, 2011). Therefore, we speculate that the protective effects of APN against HAR might also be mediated through the modulation of retinal ceramide levels. Further investigation is still ongoing to fully elucidate the contribution of lipid metabolic alterations and the ceramide pathway in HAR.

Referee #3 (Remarks):

This is a very interesting and potentially important study that combines clinical and experimental investigations to elucidate a new mechanism for the development of retinopathy of prematurity (ROP), a sight-threatening disease in premature infants. Clinically, reduced levels of adiponectin (Apn) were found to be correlated with hyperglycemia and delayed retinal vessel formation in premature infants. Experimental studies using a newly developed mouse model of hyperglycemia-associated retinopathy and 661w cell line demonstrate that depletion of Apn or activation of Apn pathway affects hyaloid vessel regression and retinal vascular formation in normal or hyperglycemic mice. Further experiments show that Apn increases retinal and photoreceptor metabolism and stimulates photoreceptor derived Pdgfb production. These findings clearly suggest a role of Apn in regulation of photoreceptor metabolism and vascular formation and therefore may serve as a potential target for ROP.

Major points:

1. A major contribution of this manuscript would be the generation of a new model for study ROP-related hyperglycemia-induced retinopathy. However, some important information appears to be missing. For example, what is the timing for the development of hyperglycemia and decrease of insulin production in this model. How about the survival rate of neonatal mice receiving STZ injection? In addition, injection of STZ from P1 - P9 only affects the deep layer but not superficial layer vascular formation. Can this phenomenon be explained by the timing for the development of hyperglycemia?

Responses: Thanks for the valuable comments. We have now examined the time course of hyperglycemia development in WT mice after STZ injections. We found that hyperglycemia was significantly induced at P8 and persisted at P10 (**Figure EV1, C**). We agree with the reviewer that the deep layer but not the superficial layer vascular formation could be partially explained by the timing of the onset of hyperglycemia, as in mouse, the superficial layer forms between P1-P10 and the deep layer starts forming at P8. At P10, when the retinal vasculature was examined, we found that the major metabolic changes were located in photoreceptors but not in other neuronal layers (**Figure 4C**) nor in the deep blood vessels (**Figure EV3, B**). As retina (in particular photoreceptors) is the most metabolically demanding tissue in the body (Wong-Riley, 2010) and photoreceptors have the highest number of mitochondria of any cell (Hoang et al, 2002), therefore, we hypothesize that the disturbance of glucose metabolism observed at P8 may lead to photoreceptor metabolic dysfunction, which in turn affect deep retinal vascular development.

Figure EV1: (C) Blood glucose levels starting from P2 to P10 (n=7-19/group, ANOVA) and serum triglyceride levels at P8 (n=5-7/group, unpaired t test) in control and HAR mice.

Unfortunately, we were not able to measure the serum insulin levels at very young ages as the mice are much too small to obtain 25µl of serum that is required according to the protocol.

Generally, we noticed that in the same litter, the STZ-induced mouse pups were weaker than the vehicle (PBS)-treated pups, which led to less capability to fight for the nursing dam's milk and in turn led to slower growth than the control pups. Therefore, we treated all mice in the same litter with either STZ or vehicle for phenotypic studies. The number of mouse pups was adjusted in PBS-treated group to achieve the similar range of body weights as the mouse body weight may affect the development of retinopathy during developmental stage (Connor et al, 2009). The pup number in the STZ-injected litter was limited to six to achieve a similar body weight range among different litters. All six mice in a litter survived at P10 and the body weight was within the 4 to 5g range.

2. It is hypothesized that hyperglycemia suppresses photoreceptor metabolism resulting in delayed retinal vascularization. However, direct evidence for this is lacking (*Apn* has no effect on hyperglycemia). Perhaps it would be more convincing to demonstrate the role of hyperglycemia in inducing photoreceptor dysfunction by supplementation of insulin to the hyperglycemic neonatal mice.

Responses: We thank the reviewer for the suggestion. We have now treated WT HAR mice with insulin to examine if hyperglycemia is the major cause. To avoid any potential drug/drug interference, we injected STZ and insulin separately. As two intraperitoneal injections per day may cause too much stress and death in neonatal mice, we started the co-treatment from P7, a day before the deep retinal vascular layer begins to form. At P10, insulin treated mice showed better deep vascular formation versus the littermate vehicle-treated mice (**Figure EV1, F**).

Figure EV1: (F) Deep retinal vascular formation in WT HAR mice with either insulin or vehicle treatments in littermates. $n=4-6$ /group. Unpaired t test.

3. While most data were derived from a large number of animals and appear to be very compelling, the quality of the deep layer vasculature seems to be suboptimal. Further, some representative images do not show the changes as indicated in the graph. For example, Figure 3C: the images on the left show fewer hyaloid vessels in *Apn*^{-/-} retinas, while the graph shows an increase in these retinas.

Responses: We have now updated the images of deep vascular network (Figure 2B, Figure EV5), as well as hyaloid vessels in Figure 3C (WT control). The quantification of the hyaloid vessels is shown below: WT control = 10; WT, HAR = 16; *Apn*^{-/-}, control = 15; *Apn*^{-/-}, HAR = 15.

4. Is AMPK activation altered by AdipoRon injection and in *Apn*^{-/-} retinas in vivo? These data would be helpful in supporting the hypothesis (Figure 5).

Responses: Thank you for the comment. We tried to do Western blots of p-AMPK and AMPK in HAR retinas. Unfortunately, unlike in 661W cells, there were multiple non-specific bands and the

technique was too unreliable to answer the question. Instead, we treated the STZ-induced mice with mouse recombinant APN (msAPN) and Compound C (AMPK inhibitor (Liu et al, 2014; McCullough et al, 2005)) or vehicle. We found that co-treatment with Compound C significantly reduced deep retinal vascular formation versus vehicle control in the presence of msAPN administration (**Figure EV3, H**). Therefore, AMPK activation is required for APN protection against hyperglycemia-induced retinal vascular changes.

Figure EV3: (H) Deep retinal vascular formation in WT HAR mice co-treated with msAPN (0.6µg/g) and Compound C (AMPK inhibitor, 2µg/g) or vehicle (DMSO, the diluting solution for Compound C) from P7 to P9. Unpaired t test.

5. It is less clear how *Apn* regulates VEGFA expression, although Figure 5 indicates with *Apn* photoreceptors produce less VEGF (+PDGFB, -VEGF)? There is no data provided in this regard.

Responses: Thanks for pointing out the missing data. We observed that although APN deficiency reduced *Vegfa* expression in the ONL layer (**Figure EV4A**), knockdown of photoreceptor-derived VEGFA did not affect normal retinal vascularization in mice (**Figure EV4, B**). We speculate that the decrease in VEGFA with APN deficiency might not contribute substantially to the retinal vascularization in the deep layer in HAR model.

Figure EV4: (A) In *Apn*^{-/-} versus WT STZ, *Vegfa* mRNA expression in retinal neuronal layers (ONL, INL, GCL) (LCM and qRT-PCR). n=3 pooled retinas/group. **(B)** Representative images of deep retinal vascular network and quantification in AAV2-hRK-Vegfa-GFP versus AAV2-hRK-GFP (control) virus subretinally injected in WT mice. Scale bar, 50µm. n=6-7 retinas/group. Unpaired t test.

We also apologize for the confusion in this figure, we have now updated the figure as shown below.

Figure 6: Schematic of the proposed role of the APN pathway in photoreceptor metabolism and deep retinal vascular development in STZ-induced hyperglycemic retinopathy. The APN pathway phosphorylates AMPK to increase mitochondrial activity, which in turn increases the production of neurovascular growth factor PDGFB. Photoreceptor-derived PDGFB (not VEGFA) promotes the deep retinal vascular network formation in hyperglycemic retinopathy.

6. The use of "photoreceptors" instead of "661w cells" is confusing and misleading, since changes in photoreceptors generally refer to results from *in vivo* studies. Furthermore, 661w cells is a cone photoreceptor cell line. The results derived from the *in vitro* study should be carefully interpreted

Responses: Thanks for pointing this out. We now use the term "661W cells" instead of the term "photoreceptors" for *in vitro* studies. We also agree with the reviewer that 661W cells are a cone photoreceptor cell line. To examine if cone function could be improved with activation of the APN pathway, cone ERGs were obtained in WT HAR mice treated with msAPN or vehicle. In the HAR model, cone function (reflected by post-receptor response V_m , because the cone response is always small in mice) was decreased versus normal control mice. The msAPN treatment significantly improved cone responses (Figure 5E). Our findings suggested that the *in vitro* data correspond to the *in vivo* data.

Figure 5: Hyperglycemia was induced by 25mg/kg STZ (i.p., daily from P2-12). The mouse pups received msAPN (0.6ug/g) or vehicle (PBS) injection daily from P7-39. At P40, cone (E) ERG were conducted. Cone ERG signals were reflected by post-receptor responses (V_m). $n=7$ to 12 per group. t test (E).

Minor points:

7. The definition of HAR should be consist in the manuscript. In the legends for Figure 2: hyperglycemia-associated retinopathy (HAR); Figure3: ...hyperglycemia-associated retinal vascularization (HAR) delay.... Which is correct?

Responses: HAR is now defined consistently as hyperglycemia-associated retinopathy. We have now updated the current manuscript.

8. Legend for Figure 4: Please specify the age of the mice used in each assay. (B): is the difference significant? (C): please clarify "In $Apn^{-/-}$ versus WT STZ" - were these mice hyperglycemic?

Responses: The mice were sacrificed and the retinas were isolated for studies at P10. In the current manuscript, we include this information. In B, we updated this figure with msAPN versus vehicle treatment in WT HAR mice, significantly increased maximal respiration was observed with msAPN treatment. (C) We have now updated it as " $Apn^{-/-}$ versus WT HAR P10 retinas", these mice are hyperglycemic.

9. Figure EV3b, c shows that normal or hyperglycemic $Apn^{-/-}$ retinas display significant reduction in superficial layer vascular formation. This information is missing in the results section. The significance and mechanism for this defect should discussed. In contrast, there is only a modest decrease in the deep layer vascular formation in $Apn^{-/-}$ retinas (Figure 3E). Is there a possible explanation for this?

Responses: We appreciate the reviewer's comment. We now state the results in the Result section as "APN deficiency inhibited superficial but not deep retinal vascularization in normal conditions at P10 (Fig. EV2B)".

We first speculated that this phenomenon might due to the persistent hyaloid vessels with APN deficiency seen at P8 (Fig. 3C, $Apn^{-/-}$, control vs. WT, control). In both mice and human eyes, the hyaloid vessels are the first to form *in utero* and regress as the retinal vessels develop (Ito & Yoshioka, 1999; Stahl et al, 2010). Therefore persistent hyaloid vessels may keep supplying the nutrients and oxygen for the retina and thus decrease the need for superficial retinal vessels. In WT HAR vs normal mice, hyaloid vessels were persistent in HAR (Fig. 3C) but there were no

significant differences in the superficial vascular network (**Figure EV1, E**). As noted, there is a sharp drop in the number counts of hyaloid vessels from P5 to P8 in mice (Rao et al, 2013). In HAR, hyperglycemia was induced from P6 to P8 (**Figure EV1, C**). The rise in blood glucose levels may interrupt the hyaloid vessel regression during this sensitive period, which leads to persistent hyaloid vessels. However, as the superficial retinal vessels cover more than 50% of the retinas at P5 and almost reach the most peripheral region at P8, the impact of hyperglycemia on the superficial vasculature may be mild. However, *Apn*^{-/-} mice, with absence of APN from conception, have a more profound change in hyaloid and even on superficial retinal vessels. Further investigation of the influence of the APN pathway on the systems controlling hyaloid vessel regression and superficial retinal vascular formation (e.g. retinal ganglion cells, RGCs (Rao et al, 2013)) are needed to better understand mechanisms. APN receptors were found in RGCs (**Fig. 3B**) although the expression is much less than in photoreceptors. In summary, APN, possibly through targeting different retinal neurons, controls retinal vascular formation. Further exploration may benefit patients with persistent hyaloid remnants (Mullner-Eidenbock et al, 2004; Saint-Geniez & D'Amore, 2004; Sharma & Biswas, 2012; Shastry, 2009).

As noted above, complete loss of APN led to persistent hyaloid vessels and suppressed superficial retinal vessel growth. The relatively mild impact of APN deficiency on deep vascular layer versus on the superficial layer is possibly due to different controls for superficial and deep retinal vessel growth. As photoreceptors are highly metabolically active and require high levels oxygen/nutrients for energy production, the formation of deep vascular vessels may be more dependent on photoreceptor metabolic needs than the superficial vessels. We do not exclude the possibility that there are other factors to compensate for the loss of APN.

We now have included the discussion in the current manuscript.

10. Symbols for indications of significance are missing in some figures.

Responses: The symbols for indications of significance are now updated.

References

- Aleidi S, Issa A, Bustanji H, Khalil M, Bustanji Y (2015) Adiponectin serum levels correlate with insulin resistance in type 2 diabetic patients. *Saudi pharmaceutical journal : SPJ : the official publication of the Saudi Pharmaceutical Society* 23: 250-256
- Chen H, Tran JT, Brush RS, Saadi A, Rahman AK, Yu M, Yasumura D, Matthes MT, Ahern K, Yang H et al (2012) Ceramide signaling in retinal degeneration. *Advances in experimental medicine and biology* 723: 553-558
- Connor KM, Krah NM, Dennison RJ, Aderman CM, Chen J, Guerin KI, Sapieha P, Stahl A, Willett KL, Smith LE (2009) Quantification of oxygen-induced retinopathy in the mouse: a model of vessel loss, vessel regrowth and pathological angiogenesis. *Nature protocols* 4: 1565-1573
- Deepa SS, Dong LQ (2009) APPL1: role in adiponectin signaling and beyond. *American journal of physiology Endocrinology and metabolism* 296: E22-36
- German OL, Miranda GE, Abraham CE, Rotstein NP (2006) Ceramide is a mediator of apoptosis in retina photoreceptors. *Investigative ophthalmology & visual science* 47: 1658-1668
- Hoang QV, Linsenmeier RA, Chung CK, Curcio CA (2002) Photoreceptor inner segments in monkey and human retina: mitochondrial density, optics, and regional variation. *Visual neuroscience* 19: 395-407
- Holland WL, Miller RA, Wang ZV, Sun K, Barth BM, Bui HH, Davis KE, Bikman BT, Halberg N, Rutkowski JM et al (2011) Receptor-mediated activation of ceramidase activity initiates the pleiotropic actions of adiponectin. *Nature medicine* 17: 55-63
- Ito M, Yoshioka M (1999) Regression of the hyaloid vessels and pupillary membrane of the mouse. *Anatomy and embryology* 200: 403-411

- Kadowaki T, Yamauchi T, Kubota N, Hara K, Ueki K (2007) Adiponectin and adiponectin receptors in obesity-linked insulin resistance. *Novartis Foundation symposium* 286: 164-176; discussion 176-182, 200-163
- Kooragayala K, Gotoh N, Cogliati T, Nellissery J, Kaden TR, French S, Balaban R, Li W, Covian R, Swaroop A (2015) Quantification of Oxygen Consumption in Retina Ex Vivo Demonstrates Limited Reserve Capacity of Photoreceptor Mitochondria. *Investigative ophthalmology & visual science* 56: 8428-8436
- LeCaire TJ, Palta M (2015) Longitudinal Analysis of Adiponectin through 20-Year Type 1 Diabetes Duration. *Journal of diabetes research* 2015: 730407
- Lindsay RS, Funahashi T, Hanson RL, Matsuzawa Y, Tanaka S, Tataranni PA, Knowler WC, Krakoff J (2002) Adiponectin and development of type 2 diabetes in the Pima Indian population. *Lancet* 360: 57-58
- Liu X, Chhipa RR, Nakano I, Dasgupta B (2014) The AMPK inhibitor compound C is a potent AMPK-independent antiangioma agent. *Molecular cancer therapeutics* 13: 596-605
- McCullough LD, Zeng Z, Li H, Landree LE, McFadden J, Ronnett GV (2005) Pharmacological inhibition of AMP-activated protein kinase provides neuroprotection in stroke. *The Journal of biological chemistry* 280: 20493-20502
- Mullner-Eidenbock A, Amon M, Moser E, Klebermass N (2004) Persistent fetal vasculature and minimal fetal vascular remnants: a frequent cause of unilateral congenital cataracts. *Ophthalmology* 111: 906-913
- Pereira RI, Snell-Bergeon JK, Erickson C, Schauer IE, Bergman BC, Rewers M, Maahs DM (2012) Adiponectin dysregulation and insulin resistance in type 1 diabetes. *The Journal of clinical endocrinology and metabolism* 97: E642-647
- Prior SL, Tang TS, Gill GV, Bain SC, Stephens JW (2011) Adiponectin, total antioxidant status, and urine albumin excretion in the low-risk "Golden Years" type 1 diabetes mellitus cohort. *Metabolism: clinical and experimental* 60: 173-179
- Rao S, Chun C, Fan J, Kofron JM, Yang MB, Hegde RS, Ferrara N, Copenhagen DR, Lang RA (2013) A direct and melanopsin-dependent fetal light response regulates mouse eye development. *Nature* 494: 243-246
- Saint-Geniez M, D'Amore PA (2004) Development and pathology of the hyaloid, choroidal and retinal vasculature. *The International journal of developmental biology* 48: 1045-1058
- Sharma V, Biswas S (2012) Vitreous hemorrhage in a premature infant with patent hyaloid artery and increased intracranial pressure. *Journal of pediatric ophthalmology and strabismus* 49 Online: e12-14
- Shastri BS (2009) Persistent hyperplastic primary vitreous: congenital malformation of the eye. *Clinical & experimental ophthalmology* 37: 884-890
- Stahl A, Connor KM, Sapieha P, Chen J, Dennison RJ, Krah NM, Seaward MR, Willett KL, Aderman CM, Guerin KI et al (2010) The mouse retina as an angiogenesis model. *Investigative ophthalmology & visual science* 51: 2813-2826
- Strettoi E, Gargini C, Novelli E, Sala G, Piano I, Gasco P, Ghidoni R (2010) Inhibition of ceramide biosynthesis preserves photoreceptor structure and function in a mouse model of retinitis pigmentosa. *Proceedings of the National Academy of Sciences of the United States of America* 107: 18706-18711
- Wong-Riley MT (2010) Energy metabolism of the visual system. *Eye and brain* 2: 99-116

Yamamoto S, Matsushita Y, Nakagawa T, Hayashi T, Noda M, Mizoue T (2014) Circulating adiponectin levels and risk of type 2 diabetes in the Japanese. *Nutrition & diabetes* 4: e130

Yilmaz MI, Sonmez A, Acikel C, Celik T, Bingol N, Pinar M, Bayraktar Z, Ozata M (2004) Adiponectin may play a part in the pathogenesis of diabetic retinopathy. *European journal of endocrinology / European Federation of Endocrine Societies* 151: 135-140

Yuan F, Liu YH, Liu FY, Peng YM, Tian JW (2014) Intraperitoneal administration of the globular adiponectin gene ameliorates diabetic nephropathy in Wistar rats. *Molecular medicine reports* 9: 2293-2300

2nd Editorial Decision

12 October 2017

Thank you for the submission of your revised manuscript to EMBO Molecular Medicine. We have now received the enclosed report from the referee who was asked to re-assess it. As you will see this referee is now fully supportive and I am pleased to inform you that we will be able to accept your manuscript.

***** Reviewer's comments *****

Referee #2 (Remarks for Author):

The authors have addressed all issues raised in a very comprehensive fashion. The answer provided and the data generated answers the vast majority of my initial concerns. Well done!

Corresponding Author Name: Lois Smith

Journal Submitted to: EMBO Mol Med

Manuscript Number: EMM_2017_07966V2